# Bio-Based Polyamide 1010 with a Halogen-Free Flame Retardant Based on Melamine–Gallic Acid Complex

**DOI:** 10.3390/polym12071482

**Published:** 2020-07-02

**Authors:** Nicoleta Levinta, Mihai Cosmin Corobea, Zina Vuluga, Cristian-Andi Nicolae, Augusta Raluca Gabor, Valentin Raditoiu, Mariana Osiac, George-Mihail Teodorescu, Mircea Teodorescu

**Affiliations:** 1National Institute for Research & Development in Chemistry and Petrochemistry-ICECHIM, 202 Spl. Independentei, 060021 Bucharest, Romania; nicoleta.levinta@yahoo.com (N.L.); zvuluga@yahoo.com (Z.V.); ca_nicolae@yahoo.com (C.-A.N.); ralucagabor@yahoo.com (A.R.G.); vraditoiu@yahoo.com (V.R.); georgemihail.teodorescu@yahoo.com (G.-M.T.); 2Faculty of Applied Chemistry and Materials Science, Politehnica University of Bucharest, 1–7 Polizu Street, s1, 01106 Bucharest, Romania; mircea.teodorescu@upb.ro; 3Department of Physics, University of Craiova, A.I. Cuza 13, 200585 Craiova, Romania; m_osiac@yahoo.com

**Keywords:** bio-based polyamide, flame retardant, composites, melamine, gallic acid

## Abstract

This work aims at developing polyamide 1010 (PA1010) composites with improved fire behavior using a halogen-free flame-retardant system based on melamine (Me) and gallic acid (GA) complexes (MA). The MA complexes were formed by hydrogen bonding, starting from 1:2, 1:1, 2:1 Me:GA molar ratios. PA1010 composites were obtained by melt mixing, followed by compression molding. MA provided a plasticizing effect on the PA1010 matrix by decreasing the glass transition temperature. The influence of MA on PA1010 chain packaging was highlighted in the X-ray diffraction patterns, mainly in the amorphous phase, but affected also the α and γ planes. This was reflected in the dynamic mechanical properties by the reduction of the storage modulus. H-bonds occurrence in MA complexes, improved the efficiency in the gaseous form during fire exposure, facilitating the gas formation and finally reflected in thermal stability, thermo-oxidative stability, LOI results, and vertical burning behavior results. PA1010 containing a higher amount of GA in the complex (MA12) displayed a limiting oxygen index (LOI) value of 33.6%, much higher when compared to neat PA1010 (25.8%). Vertical burning tests showed that all the composites can achieve the V-0 rating in contrast with neat PA1010 that has V-2 classification.

## 1. Introduction

One of the objectives of modern polymer science can be considered the production of conventional polymers from renewable sources because of the growing interest in sustainability and environmental issues. An advantage of these materials is the improvement of carbon footprint of plastic products [1]. Among the domains that have focused on the use of bio-based polymers are the automotive and electronics industry [2]. In this sense, polyamide (PA) is one of the most discussed polymers in engineering plastics field. However, despite their fair flame retardancy with respect to other polymers, their employment as substitutes for metals in several areas (for example automotive) for reducing weight and costs requires betterperformance in fire resistance. Also, the number of applications in automotive components and electrical connectors increased, requiring advanced solutions in several directions like structural, textile, insulating, or wearing parts.

As for bio-based PAs, the polymer chains can be partially or completely synthesized from bio-based components. In this regard, a variety of sustainable polyamides (PAs) from renewable resources were obtained and investigated as potential substitutes for conventional PAs, including polyamide 4.10 (PA410) [3], polyamide 5.10 (PA510) [4], polyamide 6.10 (PA610) [5], polyamide 10.10 (PA1010) [6], and polyamide 11 (PA11) [7]. Their applications are still emerging in some large sectors like automotive or electronics, while they are already active on 3D printing parts, biomedical parts and devices, packaging, and cosmetic fields [8]. In this context, the potential of the performed research can be viewed imminently applicable.PA1010 can be fully obtained from bio-based products, resulting by the polycondensation of decamethylenediamine and 1,10-decanedioic acid (sebacic acid), derived entirely from the castor bean plant (Figure 1). Among other polyamides, PA1010 shows lower both melting and degradation temperatures and storage modulus as compared to PA610, but higher than PA1012. This behavior was determined by the CH_2_/CONH ratio [9]. In general, polyamides can burn rapidly once ignited, promoting a pronounced melt-dripping behavior [10]. In this context, the incorporation of flame retardants was found as a necessity. Several studies focused on the investigation of the thermal degradation of polyamides [11,12]. Given the aliphatic nature of the bio-based PA1010, the most probable mechanism for thermal degradation could occur under the following steps:-the hydrolysis of the amide bond below the decomposition temperature (T_d_);-homolytic scission of the C–C, C–H, C–N bonds at temperatures above T_d_ or simultaneously with hydrolysis;-cyclization and homolytic scission of the products resultingfrom both reactions;-side reactions that produce carbon oxide, nitriles, ammonia, hydrocarbons.

The industrial-scale applicability of bio-based polyamides like PA1010 is still under development; therefore, there are only a few studies about the enhancement of the flame retardancy of PA1010 [13]. Another limitation of the state of the art was found also in the limited number of halogen-free flame retardants in general (especially when costs and accessible resources were considered). Ammonium polyphosphate (APP) and pentaerythritol (PER) were successfully tested for the improvement of the combustion properties of fully bio-based PA1010 [14]. Because of the lack of information, it is important to focus on the already collected results for other aliphatic PAs [15]. According to previous reports, different types of flame retardants (FRs) have been proposed for conventional PAs, including aluminumdiethylphospinate (AlPi) [16,17], APP [18], and melamine polyphosphate [19].

Melamine (Me) and its derivatives have the advantage of being a halogen-free FR, widely studied in PA6 and PA66 [20]. Lu et al. reported that PA6 composites loaded with melamine cyanurate (MC) passed the UL-94 (vertical burning test) V-0 rating, and the limiting oxygen index (LOI) value reached up to 33% when 10 wt % MC was added [21]. As for PA6, the sample containing 7 wt % MC displayed a LOI value of 31% and the UL-94 V-0 rating, compared to 23% and V2, respectively, in the case of neat PA6 [22]. In addition, melamine seems to be effective in long-chain polyamides, for example in PA11. Jin et al. introduced Me in PA11/expandable graphite (EG) composites and demonstrated the synergistic effect between the two fillers in terms of a LOI value of 32.3% when 15 wt % EG and 5 wt % Me were loaded [23]. Furthermore, Me provides the formation of hydrogen bonds with PA11, thus leading to an early decomposition and char formation of PA11. Me by itself is available and used as a flame retardant, but to improve its action, melamine derivatives—such as phosphates, cyanurates, oxalate, etc.—are usually used. The limitations occur in the environmental potential impact profile (i.e., cyanurates or organophosphates as emerging contaminants) [24]. Me is far from being a green FR, but it can be found in some insects and was proven also as a bio-based resource [25].

As already reported in the literature, melamine acts as a flame retardant by releasing inert ammonia gases/vapors, thus diluting oxygen and fuel gases [26]. Lotsch et al. investigated the thermal condensation of melamine suggesting two pathways which assume either the direct formation of melem in the first condensation step or melam is produced prior to melem formation. The melam formation was demonstrated by isolating its single crystals and it may be considered as a side product [27]. All these transformations involved ammonia elimination, an important inert gas in the FR mechanism. It was reported for PAs that the presence of MC promoted reactions such as dehydration of primary amide chain ends, scission of N-alkylamide bonds, and formation of carbodiimides [28].

Gallic acid (GA) is one of the most important compounds of the hydrolysable polyphenols group, widely distributed in a variety of plant tissues [29,30,31]. Because of its antioxidant and antimicrobial properties, GA found applications in food, cosmetics, and pharmaceutical fields [32]. The three phenolic-OH groups and one carboxyl group make GA a good candidate to form stable complexes with Me [33]. Saha et al. investigated the behavior of a new two-component hydrogel of Me and GA synthesized at different proportions of the components [34]. Some studies even reported the possibility of GA and Me mutual detection based on the formation of stable complexes, for example the calorimetric detection of Me using GA silver nanoparticles [35] or the electrochemical detection of GA using a poly(melamine) film [36]. The availability and cost-effectiveness of GA make it suitable for FR applications. GA derivatives showed their efficiency as FR in epoxy resins [37,38] and polyurethane coatings [39]. In this context, GA combination with Me may bring an improvement in the FR behavior.

To date, several studies have investigated the thermal behavior, water uptake, and mechanical properties of fully bio-based PA1010 [9,40]; however, its flame retardancy has not been studied in depth. In this regard, our study was focused on the combination of Me and GA to improve the flame retardancy of PA1010. The flame retardancy of Me and GA complexes could be explained by two processes: a) release of inert gases that dilute oxygen (ammonia, carbon dioxide, etc.) and b) formation of condensation products.

From the best of our knowledge this could be the first time that both Me and GA were used together to enhance the fire behavior of PA1010.

## 2. Materials and Methods

### 2.1. Materials

2,4,6-Triamino-1,3,5-triazine (melamine, Me) and 3,4,5-trihydroxybenzoic acid (gallic acid, GA) were purchased from Sigma Aldrich, Saint Louis, MO, USA, in white colour powder form. They were used without purification. VESTAMID^®^ Terra DS, a 100% bio-based medium viscosity semi-crystalline PA1010 (180 cm^3^/g viscosity number, 1.05 g/cm^3^ density at 23 °C), offered by EvonikRöhm GmbH, Darmstadt, Germany was used as the polymer matrix.

### 2.2. Preparation of Complexes and Composites

The Me and GA complexes (MA) were obtained by mixing the two components in different molar ratios, such as 1:1, 2:1, and 1:2. Me and GA were separately dissolved in deionized water. The two homogeneous solutions consisting in 4.3 g/L Me and 11.9 g/L GA were obtained by heating up to 40 °C under continuous stirring. The resulted GA solution was added to the Me solution and the mixture was stirred at 40 °C for 1 h. The final MA complex solution was cooled down to room temperature and then freeze-dried until a white fine powder was obtained (from the fast forming hydrogel when room temperature was attained). The resulted MA complexes were marked with MA11, MA21, and MA12, respectively.

The PA1010/MA composites were obtained under dynamic conditions by the melt mixing process. In this regard, a 30 cm^3^ mixing chamber of a Brabender Plasticorder LabStation (Brabender GmbH & Co. KG, Duisburg, Germany) was used for mixing the MA powder with the polymer matrix (PA1010). Prior to processing, PA1010 was dried at 80 °C for 6 h in a vacuum oven to remove any residual moisture. The samples were mixed at 200 °C at 60 rpm rotor speed for 7 min. The resulted mixtures were pressed in plates at a temperature of 200 °C to obtain standard test specimens for thermal, dynamic mechanical and flame-retardancy analysis. In addition, Me was added separately or in a physical mixture with GA into the polymer matrix under the same conditions, in order to be compared with the MA complexes. The specific formulations are displayed in Table 1.

### 2.3. Characterization

#### 2.3.1. Fourier Transform Infrared Spectroscopy (FTIR)

A FTIR 6300 spectrometer (JASCO Inc., Tokyo, Japan) equipped with a Golden Gate ATR (diamond crystal) was employed to analyze the functional groups of the resulted MA complexes. All the spectra were recorded in the range from 4000 to 400 cm^−1^, by accumulation of 32 scans at a resolution of 4 cm^−1^.

#### 2.3.2. Thermal Characterization

In order to investigate the thermal behavior of the samples, thermogravimetric analyses (TGA) were performed on a TA-Q5000IR (TA Instruments, New Castle, DE, USA) equipment for all the powders and PA1010 composites, by using a 100 µL platinum pan. The sample size was 3–9 mg in the case of powders and 12–20 mg for PA1010 composites. The samples were heated with 10 °C/min, up to 600 °C using nitrogen as the purge gas (40 mL/min for powder analysis and 10 mL/min for polymer composites).

The melting and crystallization behavior of PA1010 composites were determined by means of a differential scanning calorimetry (DSC) Q2000 equipment (TA Instruments, New Castle, DE, USA). In a conventional run, the heat–cool–heat (HCH) method started with an equilibration at −40 °C for 3 min. Next, the material was heated up to 240 °C, maintained for 2 min, then cooled to −40 °C, maintained for 2 min and reheated up to 240 °C, the cooling/heating rate being 10 °C/min. All measurements were performed under 5.0-grade helium with a 25 mL/min flow rate.

#### 2.3.3. Dynamic Mechanical Analysis (DMA)

DMA analyses were performed in dual cantilever mode by using a DMA Q800 (TA Instruments, New Castle, DE, USA) in order to measure the storage modulus (E′) and loss factor (tan δ) of the PA1010 composites as a function of temperature. Samples of 60 mm× 10 mm× 4 mm (length × width × thickness) size were scanned over 30–150 °C temperature range at a heating rate of 3 °C/min, at 20 μm amplitude and a frequency of 1Hz.

#### 2.3.4. X-ray Diffraction Analysis (XRD)

A Shimadzu 6000X-ray diffractometer (Shimadzu, Kyoto, Japan) with CuK_α_ -radiation K_α_ = 0.15406 nm was used to study the crystalline structure of the polyamides. Bragg’s equation was applied to calculate the interlayer spacing. The X-ray peaks were fitted to a Lorenzian curve. By the fitting procedure, the full FWHM were obtained.

#### 2.3.5. Flame Retardant Properties

The flammability of PA1010 composites was investigated by means of the limiting oxygen index (LOI) and vertical burning tests. The LOI measurement of PA1010/MA samples of 80 mm× 10mm × 4 mm (length × width × thickness) was carried out at room temperature according to ISO 4589-2:2017 by using a Stanton Redcroft FTA Flammability Unit instrument (Stanton Redcroft Ltd., London, U.K.). A typical procedure was used to measure the minimum oxygen concentration required to sustain the combustion of a specimen for 3 min with a consumption of no more than 5 cm from the length of the sample [41]. The vertical burning test was carried out on a custom flammability meter according to the ASTM D3801. The bar-shaped samples were used in vertical position and depending on the burning time and dripping behavior, the samples got a general classification: V-0 (best), V-1 (good), V-2 (drips).

## 3. Results and Discussion

### 3.1. Characterization of Me, GA, and MA Complexes

#### 3.1.1. FTIR Analysis

FTIR analysis was employed to investigate the functional groups of the resulted MA complexes (Figure 2). Melamine can provide hydrogen bonding interactions with molecules because of the H-bond donor or acceptor groups such as carboxyl and phenolic –OH groups [35].

The GA was characterized by the peak at 1697 cm^−1^, which corresponds to the C=O stretching vibration. The absorption peaks at 3342 and 3268 cm^−1^ are assigned to O–H stretch. The disappearance of these peaks in MA complex indicates that the phenolic groups and carboxylic group in GA formed H-bonding with the Me molecule [34]. GA exhibited a C–O stretching vibration at around 1246 cm^−1^, which was slightly shifted to higher wavenumbers in the MA complex because the C–O group vibration can be restricted by the hydrogen bonding with N–H protons from Me.

The characteristic signals of Me are those from 3466, 3415, and 3323 cm^−1^, which correspond to the NH_2_ symmetric stretching vibrations, whereas those at 1650, 1539, and 808 cm^−1^ are assigned to the triazine ring [42]. In the MA21 compound containing Me in a higher amount, the NH_2_ stretch appears at lower wavenumbers as compared to those of melamine.

The complex formation was clearly evidenced on the FTIR patterns since its formation can be seen on multiple absorption bands modifications, similar with the ones reported in the literature in the 1700–1300 cm^−1^ region, indicating the GA C=O and C–O band stretch modes being affected by the interaction with –NH protons of Me [34].

#### 3.1.2. Thermogravimetric Analysis

The thermal stability of Me, GA, and the resultant MA complexes were investigated by TGA and the results are shown in Figure 3. The temperature of the maximum degradation rate (T_max_) as well as the residue values at 600 °C are listed in Table 2. The onset decomposition temperature (T_on_) and T_max_ temperatures show the starting decomposition temperatures of the FR, earlier than the starting decomposition temperature of the polymer. This behavior was similar to other polyamide FR’s like Me but restricts the thermal domain of the potential applications.

As already reported by Alberti et al., the thermal behavior of GA involves a three-step decomposition process [43]. The first mass loss (6.8%) occurs in the 60–100 °C temperature range, with T_max_ at 78 °C and is due to the hydration water. By further heating the compound, it remains stable until 196°C, with no mass loss. The second degradation step occurs in the 196–265 °C temperature range with 35.5% mass loss, followed by a third decomposition stage (27.1%), in the 307–414 °C temperature range, with the maximum temperature of 339 °C. In these steps the decarboxylation of GA occurred resulting in the formation of carbon dioxide [39]. Compared to GA, Me shows one major degradation step with the T_max_ at 335 °C. By heating, melamine leads to the formation of condensation products such as: melam, melem, and melon with release of ammonia— an important flame diluent [27]. Melon is thermally stable up to 700 °C. Figure 3 show that melamine decomposes almost completely up to 600 °C, whilst GA has higher residue percentage at 600 °C. According to the results shown, GA has a higher thermal stability as compared with that of Me.

The TGA analysis was carried out to also investigate the thermal stability of MA complexes. Figure 3 shows that the DTG curve was a bit broadened in the range of 200–300 °C, indicating the simultaneous decomposition of MA into melamine, gallic acid, and subsequently into ammonia, carbon dioxide, and other volatile products described in [39]. The formation of MA complexes was evidenced in the TGA section by a distinct decomposition profile in comparison with the constituents Me and GA. This behavior was found in good agreement with the FTIR section where the specific interactions between Me and GA were highlighted by H-bonds formation. Even more, the influence of the H-bonds formation between Me and GA was reflected indirectly in the TGA residue of MA complexes. Neat GA showed potential cross-linking behavior during degradation (with a large amount of residue), but by complexing with Me the cross-linking was restricted, and the residue became similar with the one of Me (less than 1 wt %).

### 3.2. Characterization of PA1010 Composites

#### 3.2.1. FTIR Analysis

The effect of the complexes on PA1010 is, in all cases, the enhancement of the mobility of the molecular segments in the amorphous region, affecting the motion of the methylene chains and determining structural changes in the crystalline regions [44]. While the IR bands situated in the “functional groups” zone are characteristic to PA1010 amide groups, it is interesting to notice that these are insignificantly modified in regard to intensity or position in the spectra (Figure 4).

Linkages between crystalline-amorphous domains determine modifications of some IR bands. Thus, some particularities are observed in the case of the “3300” band, probably due to the changes of the hydrogen bonded N–H stretching vibrations. The band increased in intensity and became sharper when complexes were added to PA1010 due to an increase of the population of hydrogen bonds, including those established between thecomponents of the complexes and amide groups from the PA1010 chains, from melamine molecules present in the structure of the binary complexes [34].

Theband at 1180 cm^−1^ was previously found to be independent of crystallinity and it can be used as internal reference band [45]. In our study, this IR band was situated at 1166 cm^−1^ and the band specific to the amorphous phasewas found at 1122 cm^−1^. These bands were rather weak therefore an in-depth view on crystallinity should be seen in the XRD section. Only some remarks can be made at this point, first regarding the structural behavior of PA1010, which was dominated by the different mobility of the NH– and CO-side methylene segments and secondly related to the cis–trans isomerization of secondary amide groups responsible to the formation of hydrogen bonds. It was observed that in the lamellar structure the NH-side methylene chains remain in the disordered state a long time after the CO-side methylene chains are parallel arrayed and the intermolecular hydrogen bonds were formed [46]. Our results seem to be in agreement with these findings, as it was already shown in the case of methylene stretching modes in the 2920–2850 cm^−1^ region. Therefore, these bands varied in intensity, as it is seen in Figure 4 and Table 3, due to the conformational changes and lateral chain–chain interactions [47]. As a result of the perturbations created by the addition of complexes in the amorphous zones (1122cm^−1^), several small modifications were found for amide I and II region, next to CH_2_ mode in the 1418–1420 cm^−1^ region. The effect of hindering the parallel arrangement of the methylene chains from the PA1010 by the MA complexes (summarized in Table 3), was later confirmed in the structure evidenced in the XRD section, in the DMA section by the decrease of the PA1010 modulus, and the decrease in T_g_ seen in DSC section.

#### 3.2.2. Thermogravimetric Analysis

Neat PA1010 showed only one main degradation step with the onset decomposition temperature (T_on_) at 451 °C and T_max_ at 466 °C. All the PA1010 composites undergo a two-step decomposition process in nitrogen atmosphere. The results for the thermal decomposition of Me, GA, and MA complexes indicate that these compounds decompose much earlier than PA1010 decomposition. As a consequence, the resulted flame-retardant volatile radicals could emerge in the gas phase and quench the radicals that sustain PA1010 combustion [48,49].

The incorporation of Me, GA, or their combination as a physical mixture into PA1010 matrix induced a new decomposition step between 220 and 370 °C. This occurred because of the organic filler earlier degradation (as shown in Figure 5a,b). Furthermore, the pro-degradant effect of the hydroxyl group bonds of GA led to the premature degradation of PA1010 composites [38].

Figure 5 and Table 4 shows that approximately 10 wt % is lost during the first decomposition step of the PA1010/Me, PA1010/MeGA, PA1010/MA11, PA1010/MA12, and PA1010/MA21. The second decomposition step occurring at the approximately same temperature as that of PA1010, corresponds to approximately 90 wt %, which is the proportion of PA1010 in these composites. These results indicate that the applied flame retardants could act mainly as sources of ammonia and carbon dioxide and not in the pyrolysis of PA1010.

All the applied FRs (Me, GA, MA11, MA12, MA21) practically completely decompose before PA1010 even begins decomposition, indicating that the applied flame retardants do not match the pyrolysis specifics of the PA1010. Therefore, even the Me, which has the highest T_on_, among the applied FRs, decomposes completely up to 350 °C, whilst the PA1010 starts to decompose at 451 °C.

TGA experiments in air highlighted the FR influence during thermo-oxidative degradation of PA1010 in their composites (Figure 5c,d). All FRs influenced the PA1010 thermo-oxidative degradation stages on the main weight loss intervals. The most important contribution can be seen in the T_max_ (air) region with improvements of over 10 °C and lower weight loss. The FR influence can be noticed also by the presence of a third phase in the thermo-oxidative decomposition around 500 °C. This phase was more prominent for PA1010 with complexes. All these aspects suggest a certain involvement of FRs on the PA1010 stabilization during the thermo-oxidative degradation often encounter in fire behavior events.

The TGA in air underlines the involvement of the FRs degradation products in the thermo-oxidative PA1010 degradation. These products act in the PA1010 main the thermo-oxidative event (occurring in the 400–480 °C). PA1010 starts the main event in air at 420 °C (Figure 5d Deriv. Weight), were like any aliphatic polyamide, the abstraction of the hydrogen atom from N-vicinal methylene group occurs. The second maxima of PA1010 involved the propagation involved by the oxidation of the formed macroradical. The FR degradation products suppress the first event from 420 °C by enriching the PA1010 decomposition loci with gaseous species. The second event of PA1010 thermo-oxidative degradation (oxidation of the macroradical) was influenced also especially by the MA type of FR (Figure 5d Deriv.Weight), creating a different mechanism with third phase more evident near 500°C. The FR suppression of the oxidative events was in good agreement with LOI measurements.

#### 3.2.3. Differential Scanning Analysis

DSC analysis was carried out to investigate whether the addition of MA affects the crystallization behavior of PA1010. Figure 6 shows the DSC curves of PA1010 composites obtained in the cooling/second heating step, and the main results are summarized in Table 5. Bio-based PA1010 and its composites showed a double melting peak profile (two clear peaks and a shoulder barely visible) of PA1010. The shoulder could be assessed to a certain amorphous phase from PA1010 covering the two major crystalline phases. The two main peaks correspond to the following phases. For neat bio-based PA1010, the first melting peak at 189 °C corresponds to melting of the polymer fraction that crystallized previously, during cooling, while the second peak at 198 °C corresponds to the melting of the recrystallized polymer fraction during heating [40]. These two peaks could be a sign of a polymorphism effect on crystallites, meaning the presence of different crystalline forms: α, β, γ. Two basic crystal forms for PA1010 were reported: the stable triclinic α-crystal and the metastable pseudo γ-hexagonal crystal [50].

One can observe that the incorporation of Me did not change the melting temperature of bio-based PA1010. Despite this, for the PA1010 composites filled with MA, the two melting peaks shifted towards lower temperature with the increase of the Me content in the MA complex. The same behavior was observed for the crystallization temperature (T_c_). It appears that melamine in the complex has better compatibility and interaction with PA1010 by generating hydrogen bonds, in good agreement with results from FTIR and TGA section. As a consequence, a restriction of movement and rearrangement of the polymer chains occurred.

In the case of all FRs, the PA1010 crystallinity was influenced by the decrease of the normalized enthalpy for all calorimetric events. In general, FRs provide a plasticizing effect on the PA1010 matrix reflected in the reduction of the glass transitions T_g_ (Figure 6a,b). Me presence in PA1010 had an opposite effect, due to a strong involvement in the H bonding formation with polyamide molecules. This phenomenon involves a supplementary restriction of the polyamide chain mobility leading to a small increase of T_g_. This pronounced effect on T_g_ suggests a possible dispersion in all PA1010 phases, but given the rest of the results, it happens more in the amorphous one. This intimate dispersion inbetween PA1010 macromolecular chain molecules was in a good agreement with the later DMA and XRD results. The MA complexes influence both PA1010 crystalline forms by small shifts of T_m1_ and T_m2_ melting temperatures. FRs were able to reduce the PA1010 composites specific enthalpy for both melting and crystallization events.

#### 3.2.4. Dynamic Mechanical Analysis

The effect of MA on the viscoelastic behavior of PA1010 was investigated by DMA. Figure 7 shows the storage modulus (E’) and the loss factor (tan δ) of the composites as functions of temperature. The dynamic mechanical data of the analysed samples were obtained in the 30–150 °C temperature range.

It can be noticed that E’ values, corresponding to flexural mode of deformation, decrease with temperature increase due to the softening of the composites. In the temperature range from 45 to 70 °C there is a sudden drop of E’ for all samples and a lesser variation of the storage modulus at higher temperatures. This phenomenon is attributed to the relaxation of the amorphous phase and a subsequent transition of the material from glassy to rubbery state. As it can be seen, the E’ values of PA1010 composites decrease with the introduction of the complexes, demonstrating their dispersion in the polymer structure, the mobility of the polymer being influenced by both deformational and interaction characteristics.

Different behavior was noticed after the incorporation of FRs. A decrease in the stiffness of the composites was observed with the increase of the melamine content in the complex. The storage modulus at 30 °C increases from 1280 MPa for PA1010/MA21 (a decrease by 31% as compared to PA1010), to 1601 MPa for PA1010/MA12 (a decrease by 14% as compared to PA1010), with an inversion for PA1010/MA11, where E’ is 1828 MPa (as compared to 1859 MPa for PA1010).

With the increase of Me content in the MA complexes, an increase of the intensity of tan δ peak was observed as well. The highest intensity was noticed in the case of PA1010/MA21 sample, which showed an increase by 44% of the loss factor (tan δ = 0.1456) as compared to pure PA1010 (tan δ = 0.1011). It can be noticed that the T_g_ shifts to lower values with increasing Me content (from 65.76 °C for neat PA1010 to 48.27 °C for PA1010/MA21). These results are explained by the enhancement of the molecular segments mobility in the amorphous region. The increase of loss factor values and the decrease of the storage modulus highlighted the plasticizing effect of MA complex in the polymer matrix. These results are consistent with FTIR, DSC, and thermal analyses.

#### 3.2.5. X-ray Diffraction Analysis

The peak positions of the fitting procedure and the interlayer spacing was calculated by Bragg’s law and reported in Table 6. The interplanar spacings do not show any significant differences within the experimental accuracy. Two reflection planes related to α (100) (corresponding to the interchain distance in crystalline structure) located at 2θ = 20.3° and a strong reflection γ located at 2θ = 23.7° and assigned to 010/110 crystallographic planes were observed in the polyamide (Figure 8). A change in their intensities is clearly visible when the MA complexes are added in the PA1010 matrix, where the crystalline structure of α was noticed to be predominant. Under the present study, the mesomorphic β-form was not evident in the polyamide PA1010. The diffraction peak at 2θ = 8.3° might be related to the length of the chemical repeat unit, visible in the neat PA1010. In the case of PA1010 with MA complexes, the peak width was rather broad [8,51].

Table 6 displays a small variation of the peaks position in the neat PA1010 and PA1010 with MA complexes. The polyamide γ reflection peak located at 2θ = 23°shows a high position difference, since a strong influence of melamine diffraction planes was present in this range (Figure 8). Therefore, an analysis of the FWHM on γ reflection plane was rather difficult. A tiny Me peak was observed at 2θ = 22.2°, in all MA complexes. A large shift in the position of the diffraction peak located at 2θ = 8.3° was noticed. This shift can be explained by the k_α_ intensity of X-ray and by the melamine presence.

From the XRD pattern (Figure 9), the MA12 complex influenced the most (in comparison with other FRs) the neat polyamide, in both α and γ diffraction planes. The presence of MA complexes induces more packed structures of PA1010 macromolecular chains. The MA12 insertion in the PA1010 matrix was confirmed by XRD patterns. This aspect was in good agreement with the structural data highlighted in the DSC, DMA, and LOI results.

Among the MA complexes, MA12 provided the most pronounced ability for dispersion in the PA1010 matrix, but without affecting the α and γ diffraction planes (change their intensities in comparison with neat polyamide and highest spacing in the 2θ = 8.3 region). This aspect was in good agreement with FTIR data for α and γ phases.

A smaller FWHM for PA1010/MA11 and PA1010/MA21, respectively, could be explained by the insertion of MA complexes inbetween PA1010 crystalline phases (phases that cover the α and γ blocks). The results indicate that MA complexes produce different crystal bindings with the polyamide components.

The XRD shows that the physical mixture MeGA had a strong influence on the PA1010, being able to cover the polyamide similarly with MA complexes.

XRD confirmed the DSC phases and the MA involvement in PA1010 glass transition. The dispersion observed in DMA section can be attributed to the MA insertion in-between α and γ phases (as XRD data suggested). The MA12 complex provided (according to XRD pattern) the most intimate interaction with PA1010 chain molecules and this structural aspect explains the physical context (next to the chemical one) involved later in LOI and flame retardancy mechanism.

#### 3.2.6. Flame Retardant Properties

The fire behavior of PA1010 composites has been investigated with both LOI tests and vertical burning measurements (the data are summarized in Figure 10 and Table 7). The LOI value of neat PA1010 was found to be 25.8%. The addition of Me into the polymer matrix increased this value to 30.0 %. As already reported in the literature, Me leads to decreased dripping of the polymeric matrix, which means the fuel withdrawal from the flame and thus the flame retardancy of the sample [10].

The combination between Me and GA showed much better flame retarding efficiency than PA1010/Me. The Me-GA physical combination resulted in a 33.0% LOI value. Figure 10 shows that by increasing the GA amount in the MA complexes the corresponding LOI values increased, achieving a maximum at 33.6%. This enhancement could be a result of the antioxidant properties of GA [52].

Polymers with LOI > 21% are usually defined as self-extinguishing. According to Figure 10, no sample could be characterized as flammable and the composites could most likely self-extinguish. MA complexes showed promising results in the LOI analysis. The improved LOI results should be viewed as a consequence of the faster withdrawing of the PA1010/Me, PA1010/MeGA, PA1010/MA11, PA1010/MA12, and PA1010/MA21 samples from the flame, in comparison to the neat PA1010 sample. The results are in good agreement with achieving the V-0 classification in vertical burning tests and suppressing the PA1010 dripping effect. According to this test, once ignited, PA1010 produced heavy flaming drips which further ignited the cotton indicator. The after-flame times t_1_, and t_2_ for the neat PA1010 were equal to 25.0 ± 1 s and 20.0 ± 1 s, respectively. All the PA1010 composites had an after-flame time equal to 0 s (Table 7). The vertical burning measurements highlighted the flame-retardant action of melamine and gallic acid in terms of the reduced flammability of the melt-dripping, classifying these composites as V-0.

MA complexes provide a better fire-retardant effect as compared with Me and MeGA physical mixture, proven in several aspects of the study (structure, thermo-oxidative degradation, or LOI). However, the physical mixture MeGA provides close performance to MA complexes, therefore in industrial applications or in the context of balancing with other properties and applications, this choice should be considered since it displays obvious technological advantages.

Analyzing the LOI and vertical flame spread results together, it can be concluded that such FR system acts mainly in the gaseous phase. The FR performance of Me could be attributed to the higher content of ammonia in the evolved gases, while in the case of GA to carbon dioxide and carbon monoxide. The MA complexes were even more active in this direction (of gaseous form) since the H-bond occurrence can promote molecule scission and gas formation (in good agreement with earlier TGA). Because of the lower viscosity of the polymeric matrix, the gases evolved at high temperatures can diffuse through the mass at the surface, delaying the ignition of the matrix in this way.

## 4. Conclusions

Bio-based polymers, like full bio-based PA1010, are still in the emerging stages for different applications. PA1010 was proven as a suitable polymer matrix for Me and GA-based composites obtained from melt processing. Me was able to form complexes with GA and by establishing a large number of H-bonds, a distinct thermal profile was achieved. MA complexes were formed spontaneously from Me and GA solutions in water. The MA complex forms as a hydrogel in water, easily isolated by freeze-drying which converted it in a white fine powder used further for PA1010 melt processing.

PA1010 chain interactions were evidenced by pronounced crystalline phases (in XRD and DSC sections) and showed a strong hydrogen bonds interaction in the amido segments. These interactions were able to perturb the entire spatial arrangements of the macromolecules (visible also on FTIR spectra). MA complexes as FR in PA1010 influenced the polymer T_g_, which can be associated with a plasticizing effect. In the case of the dynamic mechanical properties (evidenced by DMA), the MA complexes showed their limitations, leading to a small decrease of the PA1010 composite storage modulus in comparison with neat polyamide. Future work could consider improvements in this direction by using different reinforcing agents.

PA1010 structure (by XRD) was influenced by the MA complexes as FR. PA1010 main crystalline forms found in α and γ diffraction planes, were in good agreement with the DSC and DMA data. Besides the insertion of the FR in-between PA1010 crystalline phases, the influence of MA on PA1010 chain packaging (in α and γ phases) explains the small decrease of the dynamic mechanical properties and LOI increase.

FRs based on MA complexes showed sensible improvements of PA1010 fire behavior, as evidenced by LOI and vertical flame spread tests. LOI values for PA1010/MA12 highlight the efficiency of the complexes in the gaseous form during fire exposure, since the H-bond occurrence can facilitate gas formation. Finally, improvement can be seen in the LOI value from 25.85 to 33.65 and vertical burning classification from V-2 to V-0 for the initial PA1010.

## Figures and Tables

**Figure 1 polymers-12-01482-f001:**
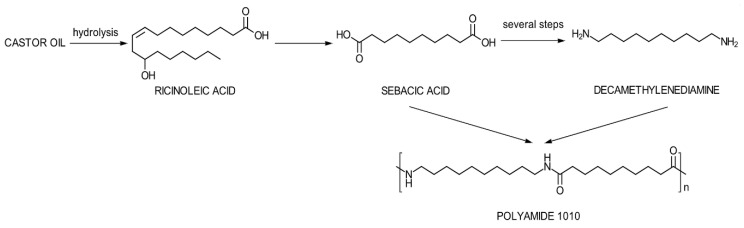
Simplified scheme for the synthesis route of bio-based PA1010 from castor oil.

**Figure 2 polymers-12-01482-f002:**
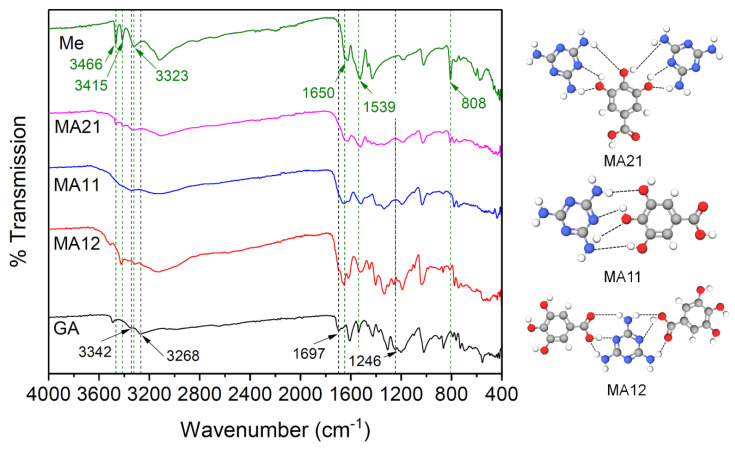
FTIR spectra of Me, GA, and MA complexes (green dash line for Me and black for GA) and the schematic illustration of possible H-bonding in MA complexes.

**Figure 3 polymers-12-01482-f003:**
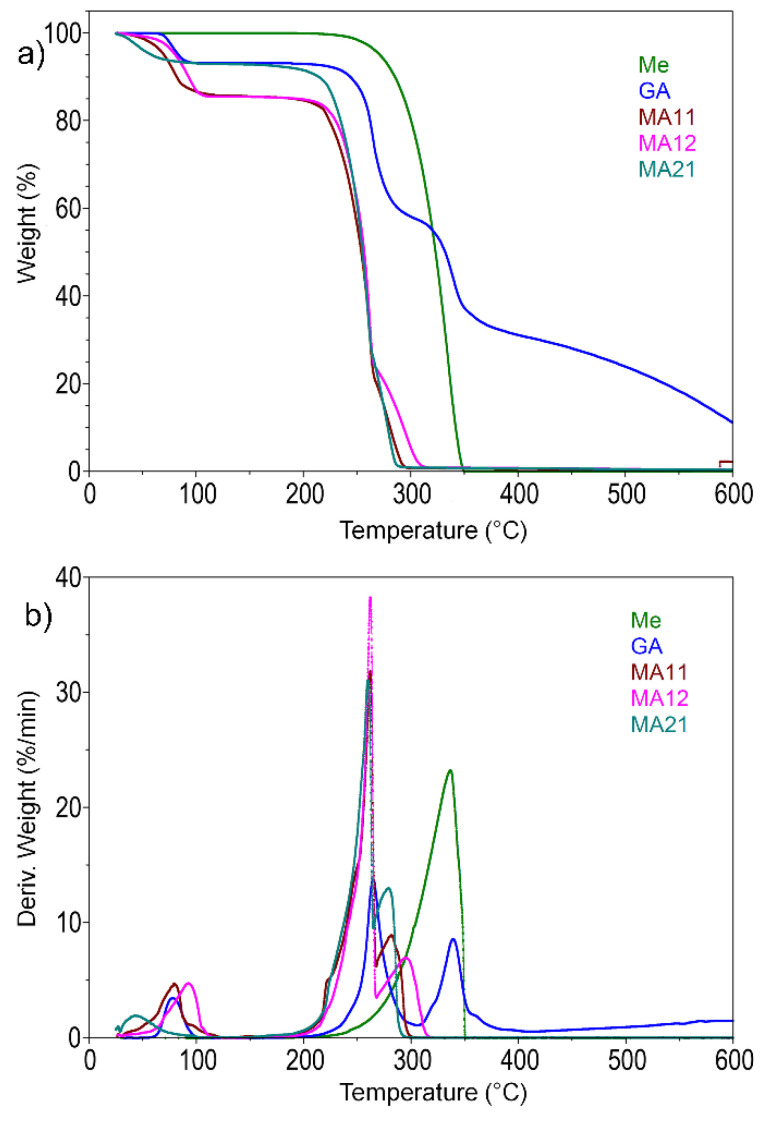
Thermogravimetric curves (TG) (**a**) and their derivatives (DTG) (**b**) for Me, GA, and MA complexes.

**Figure 4 polymers-12-01482-f004:**
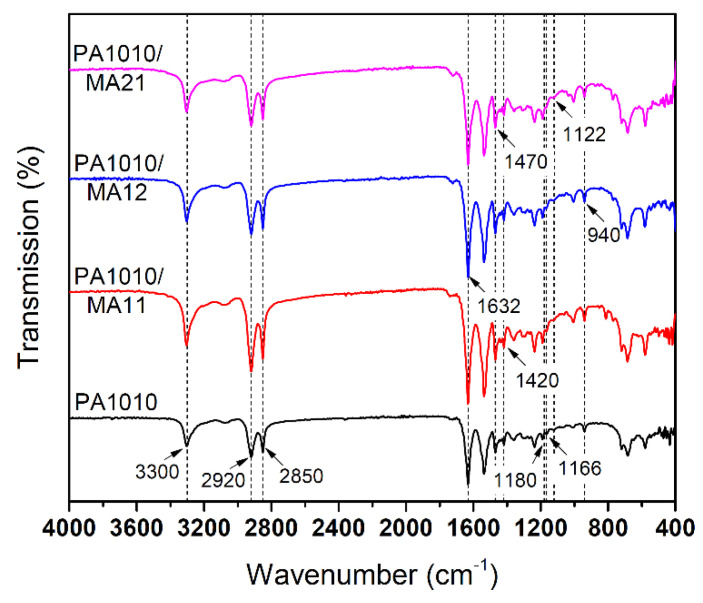
FTIR spectra of neat PA1010 and PA1010 filled with MA complexes.

**Figure 5 polymers-12-01482-f005:**
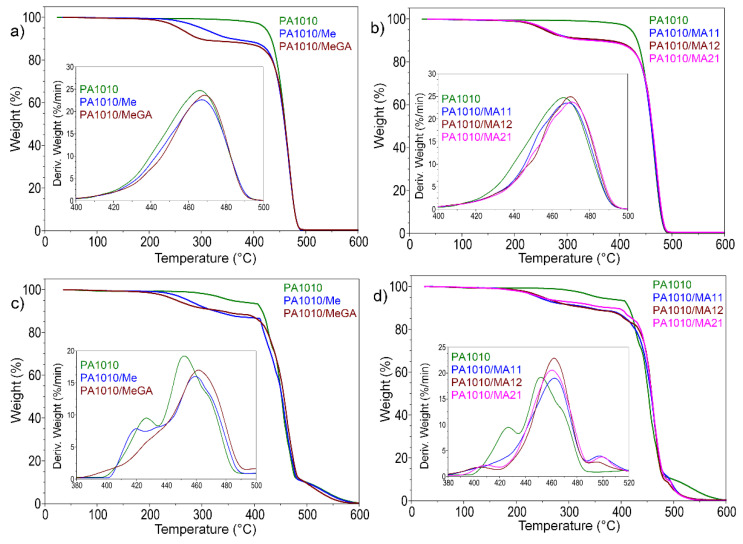
Thermogravimetric curves (TG) and their derivatives (DTG) for PA1010 composites in nitrogen (**a**,**b**) and in air (**c**,**d**).

**Figure 6 polymers-12-01482-f006:**
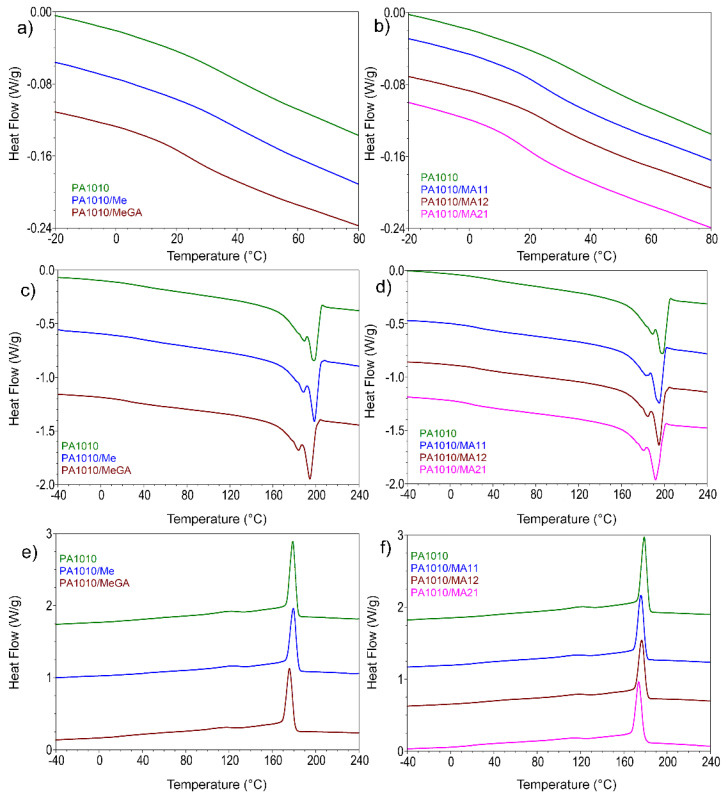
DSC thermograms of PA1010 and its composites with Me/MeGA/MA: second heating curves: (**a**,**b**) from −20 to 80 °C; (**c**,**d**) from −40 to 240 °C; (**e**,**f**) cooling curves.

**Figure 7 polymers-12-01482-f007:**
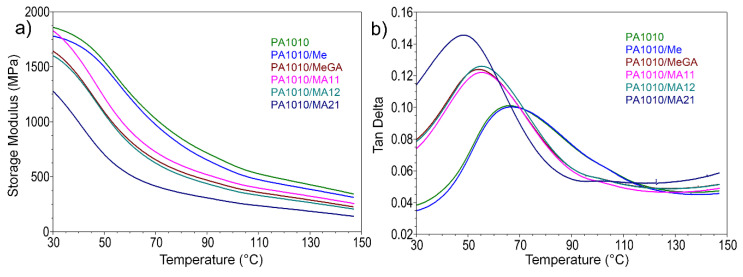
Storage modulus (**a**) and Tan δ (**b**) of PA1010 composites vs. temperature.

**Figure 8 polymers-12-01482-f008:**
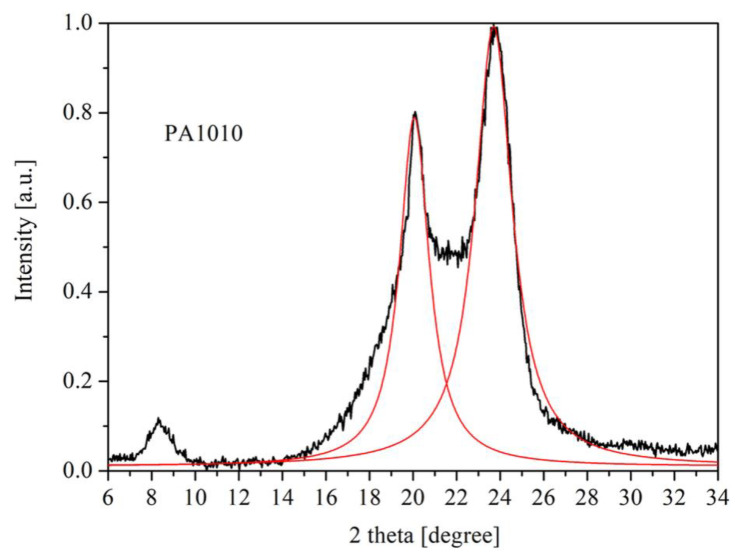
XRD pattern of neat PA1010. The red solid lines are the Lorenz fitting of α and γ diffraction planes.

**Figure 9 polymers-12-01482-f009:**
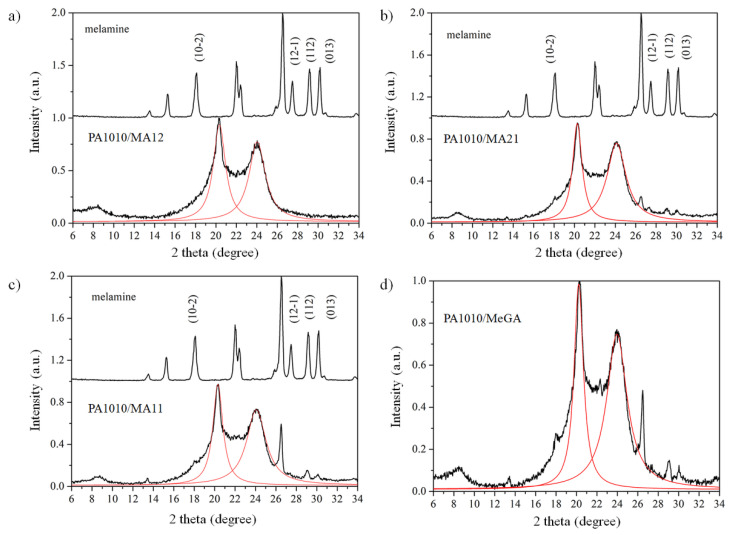
XRD pattern of PA1010 with MA complexes. (**a**) polyamide with MA12; (**b**) polyamide with MA21; (**c**) polyamide with MA11; (**d**) polyamide with MeGA. The red solid lines represent the Lorenz fitting of α and γ diffraction planes.

**Figure 10 polymers-12-01482-f010:**
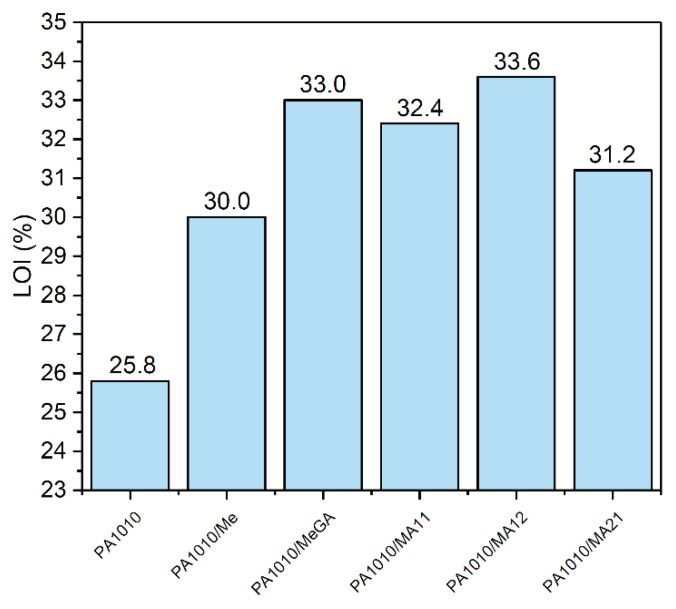
LOI values of PA1010 composites.

**Table 1 polymers-12-01482-t001:** PA1010 composites formulations.

Samples	PA1010 (wt %)	Me (wt %)	GA (wt %)	MA11 (wt %)	MA12 (wt %)	MA21 (wt %)
PA1010	100	-	-	-	-	-
PA1010/Me	90	10	-	-	-	-
PA1010/MeGA	90	5	5	-	-	-
PA1010/MA11	90	-	-	10	-	-
PA1010/MA12	90	-	-	-	10	-
PA1010/MA21	90	-	-	-	-	10

**Table 2 polymers-12-01482-t002:** TG/DTG data corresponding to Me, GA, and MA complexes.

Sample	T_on_	T_max_	Residue at T_max_	Residue at 600 °C
	(°C)	°C	%	%
Me	323.1	335.5	24.37	0.16
GA	58.5 196.7 318.1	78.1 264.9 339.3	96.89 76.17 44.51	11.23
MA11	64.2 216.0 252.8 272.0	79.4 222.8 260.7 280.8	91.59 82.94 35.00 10.30	0.5
MA12	78.9 256.9 274.0	92.9 262.1 295.4	90.1 34.2 8.8	0.4
MA21	30.8 220.6 253.8 266.0	42.7 223.9 260.0 279.0	97.6 87.4 36.8 9.8	0.4

**Table 3 polymers-12-01482-t003:** Assignments of the FTIR bands for PA1010 and PA1010/MA12 composites.

PA1010	PA1010/MA12	Band Assignment	
3303	3303	N–H stretching vibration	General
3075	3075	N–H stretch and amide II overtone	General
2919	2918	CH_2_ asymmetric stretching	General
2850	2851	CH_2_ symmetric stretching	General
1738	1738	C=O stretch (ester)	General
1632	1633	Amide I band (C=O stretching)	General
1538	1539	Amide II band (N–H in-plane bending coupled with C–N and C–O stretch)	General
1470	1467	CH_2_ scissoring not adjacent to the amide group	α-structure
1418	1420	CH_2_ scissoring	α-structure
1435	1435	CH_2_ scissors vibration	γ-structure
1359	1359	CH_2_ twist-wagging	γ-structure
1237	1236	CH_2_ twist-wagging	γ-structure
1190	1189	CH_2_ twist-wagging	α-structure
940	941	Vibration of the N-vicinal CH_2_ group coupled amide III “crystalline band”	α-structure
720	718	Rocking mode of CH_2_	α-structure γ-structure

**Table 4 polymers-12-01482-t004:** TG/DTG data corresponding to PA1010/Me/MeGA/MA composites in nitrogen and air.

Sample	Nitrogen				Air			
	T_on_	T_max_	Residue at T_max_	Residue at 600 °C	T_on_	T_max_	Residue at T_max_	Residue at 600 °C
	(°C)	(°C)	(%)	(%)	(°C)	(°C)	(%)	(%)
PA1010	451.4	466.1	36.56	0.16	421.8 446.7	426.7 451.8	80.66 50.42	0.38
PA1010/Me	272.4 457.1	324.1 467.9	92.63 31.16	0.19	417.1 452.1	419.9 459.4	78.90 38.94	0.29
PA1010/MeGA	231.4 455.8	273.7 468.6	92.72 31.32	0.51	450.7 494.4	461.7 522.5	41.16 5.964	0.42
PA1010/MA11	223.1 459.1	261.5 468.6	94.98 32.17	0.38	450.1 489.1	462.3 497.5	39.15 7.79	0.58
PA1010/MA12	232.7 464.5	257.6 469.8	94.89 32.65	0.40	454.8 488.4	462.3 496.2	40.86 6.42	0.54
PA1010/MA21	230.0 466.7	270.5 470.5	94.17 32.00	0.60	449.6 489.6	460.7 499.7	44.65 6.10	0.25

**Table 5 polymers-12-01482-t005:** Main thermal properties obtained from the DSC curves in terms of glass transition temperature (T_g_), melting temperature (T_m_), normalized enthalpy of melting (ΔH_m_), crystallization temperature (T_c_), and normalized enthalpy of crystallization (ΔH_c_).

Sample	T_g_	T_m1_/T_m2_	ΔH_m_	ΔH_m1_/H_m2_	T_c_	ΔH_c_
	°C	°C	J/g	J/g	°C	J/g
PA1010	36.6	189.3/198.2	75.94	47.25/28.69	178.9	52.94
PA1010/Me	37.5	188.5/198.6	66.26	38.82/27.44	179.4	46.85
PA1010/MeGA	23.9	184.0/194.4	65.74	36.65/29.09	175.9	46.63
PA1010/MA11	24.0	183.2/195.4	68.21	37.71/30.51	176.0	43.44
PA1010/MA12	25.9	184.8/195.1	63.18	35.06/28.12	176.7	40.50
PA1010/MA21	18.7	180.9/192.0	68.46	35.05/33.42	173.9	41.76

**Table 6 polymers-12-01482-t006:** XRD diffraction planes and the lattice spacing for PA1010 and PA1010 with MA complex under investigation.

Samples	Positions (degree)			Full Width at the Half Maximum (FWHM)			Spacing/nm		
PA1010	8.37	20.06	23.7	1.57	1.65	2.08	1.058	0.449	0.383
PA1010/MeGA	8.44	20.21	23.8	2.38	1.15	2.42	1.049	0.446	0.382
PA1010/MA11	8.50	20.32	24.08	1.81	1.19	2.30	1.042	0.444	0.377
PA1010/MA12	8.15	20.34	24.04	2.74	1.45	1.95	1.088	0.443	0.378
PA1010/MA21	8.50	20.28	24.12	2.36	1.11	2.25	1.042	0.444	0.377

**Table 7 polymers-12-01482-t007:** Results and classifications of the vertical burning measurements performed on PA1010 and its composites.

Sample	t_1_	t_2_	Cotton Indicator Ignition	Classification
	(s)	(s)		
PA1010	25.0 ± 1	20.0 ± 1	Yes	V-2
PA1010/Me	0	0	No	V-0
PA1010/MeGA	0	0	No	V-0
PA1010/MA11	0	0	No	V-0
PA1010/MA12	0	0	No	V-0
PA1010/MA21	0	0	No	V-0

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
