# Peer review of "Bio-Based Polyamide 1010 with a Halogen-Free Flame Retardant Based on Melamine–Gallic Acid Complex"

_polymers, 2020, doi:10.3390/polym12071482_

Round 1

Reviewer 1 Report

This work aimed to improve the flame retardancy of PA1010 through incorporating the complexes or melamine gallate originated from melamine and gallic acid. In fact, regarding the high-temperature application of PA1010, it is not suitable to adopt melamine gallate to modify PA1010 because of its relatively low themal stability, which induced the themal decomposition of PA1010 during the melt mixing process. Meanwhile, the characterization of the constitution of MA11, MA21 and MA12 and the binding force of Me and GA were not clear. In addition, this work didnot present the resutls of UL-94 and cone calorimeter test.

Author Response

Comments and Suggestions for Authors

This work aimed to improve the flame retardancy of PA1010 through incorporating the complexes or melamine gallate originated from melamine and gallic acid. In fact, regarding the high-temperature application of PA1010, it is not suitable to adopt melamine gallate to modify PA1010 because of its relatively low themal stability, which induced the themal decomposition of PA1010 during the melt mixing process. Meanwhile, the characterization of the constitution of MA11, MA21 and MA12 and the binding force of Me and GA were not clear. In addition, this work didnot present the resutls of UL-94 and cone calorimeter test.

Thank you for the comments and the time spent reviewing of our manuscript. The English was revised, and we added the changes. Indeed the melamine gallate comes from a hydrogel and the residual water, can be found in large amounts with negative effects on processing stage. However, the processing conditions were very close to the decomposition limit in order to promote an intimate mixing between phases. We reshaped the Table with decomposition temperatures in order to highlight the decompositions above 200 ºC.Melamine which is a regular flame retardant for polyamides, induces the same effect in PA1010 (Figure5).  Gallic acid as well is lowering to much the thermal stability, however the complex and even the physical mixture was found as a fair compromise. We would like also to highlight that a lot of existing flame retardants (melamine alone also) induce in polyamides a decrease of thermal stability [1]. We added more at the FTIR figure to highlight the constitution of MA11, MA21 and MA12 and the binding force of Me and GA. We highlighted in several places of the revised manuscript the H-bond involvment in MA complexes. Unfortunately, we could not perform the cone calorimeter test because we do not have the necessary equipment in our institute. In fact, we are looking for collaborations in this direction. After your comments, we performed additional work and adapted an experimental set-up according to ASTM D3801 for UL-94 measurements.

  1. Liu, Y., et al., Preparation of polyamide resin-encapsulated melamine cyanurate/melamine phosphate composite flame retardants and the fire-resistance to glass fiber-reinforced polyamide 6. Journal of Applied Polymer Science, 2006. 102(2): p. 1773-1779.

Thank you for all the valuable comments and we hope you will find our modifications satisfactory.

Best Regards

Mihai Corobea

Reviewer 2 Report

As a reviewer, I suggest that the paper entitled “Full bio-based Polyamide 1010 with a halogen-free flame retardant based on Melamine-Gallic Acid complex” should be reconsidered for publication in the Polymers after changes made according to the major revision.

I propose the following suggestions to the authors, which I believe could increase the impact of this work:

  1. First, I would like to congratulate to authors for their work.
  1. Abstract: The abstract highlights the main objective, but I believe that later one should be preceded by the addressing the flammability problem for PA1010. Additionally, the abstract will be much more informative if authors include the information regarding the used preparation methods instead of listing the characterization methods, as well as if include the explanations for the results (the main conclusions) instead listing them without connecting them into coherent conclusions. E.g. Why despite the non-improved thermal stability, LOI increased? How MA influenced PA1010 packaging and what caused decrease of the elasticity?
  1. Introduction: Did you mean “the environmental aspects” instead of “the life-cycle assessment”?
  1. Introduction: The state of the art regarding the mechanism of thermal decomposition of PA1010 and flame retardant action of the applied compounds (gas phase, condensed phase) in the published literature should be provided in the introduction.
  1. Introduction: The parts of the section 3.3. Thermal degradation mechanism representing the summary of the already published results regarding the mechanism of the thermal decomposition of PA1010 and flame retardancy mechanisms should be shortened and move to the introduction part, and should the followed by the hypothesis regarding the use of the combination of Me and GA for improving the flame retardancy of PA1010.
  1. Materials and Methods: The sample PA10101/GA is missing in the Table 1.
  2. Results and Discussion (FTIR – Me, GA and MA complexes): I suggest to authors to mark (with doted lines) the characteristic bands for GA and ME in accordance to the discussed results, indicating the formation of hydrogen bonds between carboxylic group of gallic acid and amino groups of melamine in MA21, MA11 and MA12.
  3. Results and Discussion (Thermogravimetry – Me, GA and MA complexes): I suggest using the term “thermal decomposition” instead of “thermal degradation”.
  4. Results and Discussion (Thermogravimetry – Me, GA and MA complexes): The thermogravimetry results for GA are missing in the Table 2. Also, the results for the ME in Table 2 include T2 and Tmax4 temperatures, whilst ME has only one decomposition step. In general, Table 2 is (at least for me) a little bit confusing, as there are no provided explanations for the used codes. I believe that this table would be much more informative and helpful regarding the improving the corresponding discussion, if the authors provide results for the Tonset and Tmax temperatures, presented as Tmax for ME, and for the GA and MA complexes the Tmax should be devided into Tmax1, Tmax2 and Tmax3. Each of these Tmax should be followed by the Residue percentage at these specific temperatures.
  5. Results and Discussion (Thermogravimetry – Me, GA and MA complexes): I suggest putting the thermograms corresponding to the GA and ME, and MA21, MA11 and MA12 on the same graph (the thickness of the lines should be reduced). Also, discussion regarding these results should be corrected. Namely, according to the results presented in Fig. 3, it can be observed that melamine decomposes almost completely up to 600 °C, whilst GA has higher residue percentage at 600 °C. According to the presented results, GA has higher thermal stability as compared with that of the ME, and not the opposite as written in the manuscript.
  6. Results and Discussion (Thermogravimetry – Me, GA and MA complexes): Similarly, as provided for the melamine, the explanation regarding the processes occurring during the thermal decomposition for the GA should be included in the manuscript. These explanations can be further used for explaining the thermal stability of the MA complexes.
  7. Results and Discussion (Thermogravimetry – Me, GA and MA complexes): Since that the thermogravimetry results for the GA are missing in the Table 2 and thermograms are not presented in one graph it is difficult to discuss the influence of the melamine on the thermal stability of gallic acid (or vice versa). Apparently, the weight residue percentages at 600 °C for the Me and MA complexes are significantly lower in comparison to that of the GA, and in general these results should be supported by the explanation.
  8. Results and Discussion: The subtitle “3.2. Characterization of Me, GA and MA complexes” should be corrected.
  9. Results and Discussion (FTIR – PA1010 and composites): This discussion should be shortened and the discussion should be supported by the commented results. The commented wavelengths should be marked in the spectra in Figure 4.
  10. Results and Discussion (Thermogravimetry – PA1010 and composites): I suggest to authors to present the thermogravimetry results in the Table 4 in the same manner as I suggested for the Table 2.
  11. Results and Discussion (Thermogravimetry – PA1010 and composites): This discussion should be corrected as the included FRs did not increase the thermal stability of PA1010. Namely, the results for the thermal decomposition of ME, GA and MA complexes indicates that these compounds decompose at temperatures below the temperatures at which PA1010 even start with the decomposition. E.g. PA1010 start to decompose at 451 °, whilst the MA complexes decompose at much lower temperatures below 350 °C. This would mean that the applied FRs practically completely decompose before PA1010 even start with the decomposition. This is also visible from the Figure 5, as approximately 10 wt. % is loss during the first decomposition step of the PA1010/Me, PA1010/MeGA, PA1010/MA11, PA1010/MA12 and PA1010/MA21. The second decomposition step occurring at the approximately same temperature as that of the PA1010, corresponds to approximately 90 wt. %, which is the portion of the PA1010 in these composites. These results indicate that the applied flame retardants do not match the pyrolysis specifics of the PA1010. The authors can refer for more explanation of the relation between polymer-flame retardant in:

Vasiljević, J., Čolović, M., Jerman, I.;, Simončič, B., Demšar, A., Samaki, Y., Šobak, M., Šest, E., Golja, B., Leskovšek, M., Bukošek, V., Medved, J., Barbalini, M., Malucelli, G., Bolka, S., 2019. In situ prepared polyamide 6/DOPO-derivative nanocomposite for melt-spinning of flame retardant textile filaments. Polym. Degrad. Stab. 166, 50−59. https://doi.org/10.1016/j.polymdegradstab.2019.05.011.

Vasiljević, J., Čolović, M., Čelan Korošin, N., Šobak, M., Štirn, Ž., Jerman, I., 2020. Effect of different flame-retardant bridged DOPO derivatives on properties of in situ produced fiber-forming polyamide 6. Polymers. 12, 657. https://doi.org/10.3390/polym12030657.

  1. Results and Discussion (Limiting Oxygen Index): Considering the results of the thermogravimetry measurements and the dripping phenomenon, the imporved LOI results should be discussed as a consequence of the faster withdrawing of the PA1010/Me, PA1010/MeGA, PA1010/MA11, PA1010/MA12 and PA1010/MA21 samples from the flame in comparison to that of the PA1010 sample. These results should not be misunderstood with improved flame retardancy, but in a manner that for these kinds of samples the additional flame retardancy tests should be applied in order to obtain a fair characterization of the sample burning behaviour.
  2. I suggest to authors to correct the discussion in accordance to the presente results and to avoid using of the assumptions, which are not supported by the results, e.g. “The increase in the TGA residue could suggest the presence of a crosslinking phase. The behaviour was more pronounced when GA and Me were together either in a physical mixture or in the MA complex structure” – According to the results, there was no significant increase in the TGA residue and there is no proof for the crosslinking occurring in the condensed phase.
  3. The references should be included for the Figure 11, and the theoretical assumptions regarding the possible synergistic action between ME and GA should be commented shortly in the introduction part, whilst synergism should not be commented for the achieved results as it is not supported by results presented in this manuscript.
  4. I believe that DSC, DMA, XRD results should be commented prior the thermogravimetry and LOI results.
  5. Conclusions: Information already appearing in the introduction should not be repeated in the conclusion part. Some parts of the conclusions are not supported by the presented results! Please, correct this in order to provide the fair and coherent concluding information.

Author Response

Comments and Suggestions for Authors

As a reviewer, I suggest that the paper entitled “Full bio-based Polyamide 1010 with a halogen-free flame retardant based on Melamine-Gallic Acid complex” should be reconsidered for publication in the Polymers after changes made according to the major revision.

Thank you for the comments and the time spent for the review of our manuscript.

I propose the following suggestions to the authors, which I believe could increase the impact of this work:

  1. First, I would like to congratulate to authors for their work.

Thank you for the appreciation of our effort.

  1. Abstract: The abstract highlights the main objective, but I believe that later one should be preceded by the addressing the flammability problem for PA1010. Additionally, the abstract will be much more informative if authors include the information regarding the used preparation methods instead of listing the characterization methods, as well as if include the explanations for the results (the main conclusions) instead listing them without connecting them into coherent conclusions. E.g. Why despite the non-improved thermal stability, LOI increased? How MA influenced PA1010 packaging and what caused decrease of the elasticity?

 Given also the reviewer 3 observations we modified with: This work aims at developing polyamide 1010 (PA1010) composites with improved fire behaviour using a halogen-free flame retardant system based on melamine (Me) and gallic acid (GA) complexes (MA). The MA complexes were formed by hydrogen bonding, starting from 1:2, 1:1, 2:1 Me:GA molar ratios. Since polyamide materials replace in some applications metal materials, improving the flame retardancy could expand the applications of bio-based polyamides. PA1010 composites were obtained by simple methods with industrial relevance (melt mixing, followed by compression moulding). The addition of MA fire retardants provided a plasticizing effect on the PA1010 matrix, reflected in the decrease of the glass transition temperature (Tg). The influence of MA on PA1010 chain packaging was highlighted in the X-ray diffraction(XRD) patterns. The PA1010 MA composites showed a decrease of storage modulus in the dynamic mechanical analysis (DMA). PA1010 containing a higher amount of GA in the complex (MA12) displayed a limiting oxygen index(LOI) value of 33.6 %, much higher when compared to neat PA1010 (25.8 %). Vertical burning tests (UL-94) showed that all the composites can achieve the UL-94 V-0 rating in contrast with neat PA1010 that has V-2 classification.

  1. Introduction: Did you mean “the environmental aspects” instead of “the life-cycle assessment”?

 We modified accordingly to the environmental aspects.

  1. Introduction: The state of the art regarding the mechanism of thermal decomposition of PA1010 and flame retardant action of the applied compounds (gas phase, condensed phase) in the published literature should be provided in the introduction.

We modified accordingly and we added in the introduction

  1. Introduction: The parts of the section 3.3. Thermal degradation mechanism representing the summary of the already published results regarding the mechanism of the thermal decomposition of PA1010 and flame retardancy mechanisms should be shortened and move to the introduction part, and should the followed by the hypothesis regarding the use of the combination of Me and GA for improving the flame retardancy of PA1010.

 We modified accordingly.

  1. Materials and Methods: The sample PA10101/GA is missing in the Table 1.

Indeed, initially we thought to use also PA1010/ GA to explain the thermal stability influence. But our materials amounts were limited and moreover we found a reference involving a pro-degradant effect of the hydroxyl group bonds of gallic acid [2], therefore we did not consider it. Now we cannot add more experiments since the bio-PA1010 was limited. 

  1. Karaseva, V., et al., New Biosourced Flame Retardant Agents Based on Gallic and Ellagic Acids for Epoxy Resins. Molecules, 2019. 24(23).
  1. Results and Discussion (FTIR – Me, GA and MA complexes): I suggest to authors to mark (with doted lines) the characteristic bands for GA and ME in accordance to the discussed results, indicating the formation of hydrogen bonds between carboxylic group of gallic acid and amino groups of melamine in MA21, MA11 and MA12.

We modified accordingly and we added the structures near the figure for a more suggestive image of the H bonds.

  1. Results and Discussion (Thermogravimetry – Me, GA and MA complexes): I suggest using the term “thermal decomposition” instead of “thermal degradation”.

We modified accordingly to thermal decomposition

  1. Results and Discussion (Thermogravimetry – Me, GA and MA complexes): The thermogravimetry results for GA are missing in the Table 2. Also, the results for the ME in Table 2 include T2 and Tmax4 temperatures, whilst ME has only one decomposition step. In general, Table 2 is (at least for me) a little bit confusing, as there are no provided explanations for the used codes. I believe that this table would be much more informative and helpful regarding the improving the corresponding discussion, if the authors provide results for the Tonset and Tmax temperatures, presented as Tmax for ME, and for the GA and MA complexes the Tmax should be devided into Tmax1, Tmax2 and Tmax3. Each of these Tmax should be followed by the Residue percentage at these specific temperatures.

We modified accordingly - we restructured it in the mentioned terms.

  1. Results and Discussion (Thermogravimetry – Me, GA and MA complexes): I suggest putting the thermograms corresponding to the GA and ME, and MA21, MA11 and MA12 on the same graph (the thickness of the lines should be reduced). Also, discussion regarding these results should be corrected. Namely, according to the results presented in Fig. 3, it can be observed that melamine decomposes almost completely up to 600 °C, whilst GA has higher residue percentage at 600 °C. According to the presented results, GA has higher thermal stability as compared with that of the ME, and not the opposite as written in the manuscript.

We modified accordingly- we restructured it in the mentioned terms.

  1. Results and Discussion (Thermogravimetry – Me, GA and MA complexes): Similarly, as provided for the melamine, the explanation regarding the processes occurring during the thermal decomposition for the GA should be included in the manuscript. These explanations can be further used for explaining the thermal stability of the MA complexes.

Indeed -we modified accordingly.

  1. Results and Discussion (Thermogravimetry – Me, GA and MA complexes): Since that the thermogravimetry results for the GA are missing in the Table 2 and thermograms are not presented in one graph it is difficult to discuss the influence of the melamine on the thermal stability of gallic acid (or vice versa). Apparently, the weight residue percentages at 600 °C for the Me and MA complexes are significantly lower in comparison to that of the GA, and in general these results should be supported by the explanation.

Indeed -we modified accordingly.

  1. Results and Discussion: The subtitle “3.2. Characterization of Me, GA and MA complexes” should be corrected.

yes, we modified accordingly.

  1. Results and Discussion (FTIR – PA1010 and composites): This discussion should be shortened, and the discussion should be supported by the commented results. The commented wavelengths should be marked in the spectra in Figure 4.

We corrected.

  1. Results and Discussion (Thermogravimetry – PA1010 and composites): I suggest to authors to present the thermogravimetry results in the Table 4 in the same manner as I suggested for the Table 2

the suggestion was very good and we modified accordingly.

  1. Results and Discussion (Thermogravimetry – PA1010 and composites): This discussion should be corrected as the included FRs did not increase the thermal stability of PA1010. Namely, the results for the thermal decomposition of ME, GA and MA complexes indicates that these compounds decompose at temperatures below the temperatures at which PA1010 even start with the decomposition. E.g. PA1010 start to decompose at 451 °, whilst the MA complexes decompose at much lower temperatures below 350 °C. This would mean that the applied FRs practically completely decompose before PA1010 even start with the decomposition. This is also visible from the Figure 5, as approximately 10 wt. % is loss during the first decomposition step of the PA1010/Me, PA1010/MeGA, PA1010/MA11, PA1010/MA12 and PA1010/MA21. The second decomposition step occurring at the approximately same temperature as that of the PA1010, corresponds to approximately 90 wt. %, which is the portion of the PA1010 in these composites. These results indicate that the applied flame retardants do not match the pyrolysis specifics of the PA1010. The authors can refer for more explanation of the relation between polymer-flame retardant in:

 Vasiljević, J., Čolović, M., Jerman, I.;, Simončič, B., Demšar, A., Samaki, Y., Šobak, M., Šest, E., Golja, B., Leskovšek, M., Bukošek, V., Medved, J., Barbalini, M., Malucelli, G., Bolka, S., 2019. In situ prepared polyamide 6/DOPO-derivative nanocomposite for melt-spinning of flame retardant textile filaments. Polym. Degrad. Stab. 166, 50−59. https://doi.org/10.1016/j.polymdegradstab.2019.05.011.

 Vasiljević, J., Čolović, M., ČelanKorošin, N., Šobak, M., Štirn, Ž., Jerman, I., 2020. Effect of different flame-retardant bridged DOPO derivatives on properties of in situ produced fiber-forming polyamide 6. Polymers. 12, 657. https://doi.org/10.3390/polym12030657.

We reshaped the discussion and we founded the references very useful, therefore we modified accordingly.

  1. Results and Discussion (Limiting Oxygen Index): Considering the results of the thermogravimetry measurements and the dripping phenomenon, the improved LOI results should be discussed as a consequence of the faster withdrawing of the PA1010/Me, PA1010/MeGA, PA1010/MA11, PA1010/MA12 and PA1010/MA21 samples from the flame in comparison to that of the PA1010 sample. These results should not be misunderstood with improved flame retardancy, but in a manner that for these kinds of samples the additional flame retardancy tests should be applied in order to obtain a fair characterization of the sample burning behaviour.

We modified accordingly in the LOI section.

  1. I suggest to authors to correct the discussion in accordance to the presente results and to avoid using of the assumptions, which are not supported by the results, e.g. “The increase in the TGA residue could suggest the presence of a crosslinking phase. The behaviour was more pronounced when GA and Me were together either in a physical mixture or in the MA complex structure” – According to the results, there was no significant increase in the TGA residue and there is no proof for the crosslinking occurring in the condensed phase.

Yes indeed we corrected and we modified accordingly.

  1. The references should be included for the Figure 11, and the theoretical assumptions regarding the possible synergistic action between ME and GA should be commented shortly in the introduction part, whilst synergism should not be commented for the achieved results as it is not supported by results presented in this manuscript.

Correct and we modified accordingly.

  1. I believe that DSC, DMA, XRD results should be commented prior the thermogravimetry and LOI results.

We modified accordingly, less TGA. Unfortunately, we founded less productive for our manuscript since both TGA and DSC belong to thermal characterisation, hoping you will understand us.   

  1. Conclusions: Information already appearing in the introduction should not be repeated in the conclusion part. Some parts of the conclusions are not supported by the presented results! Please, correct this in order to provide the fair and coherent concluding information.

We reshaped the entire section also based on the comments of reviewer 3 -we modified accordingly.

Thank you for all the valuable comments and we hope you will find our modifications satisfactory.

Best Regards

Mihai Corobea

Reviewer 3 Report

The paper “Full bio-based Polyamide 1010 with a halogen-free flame retardant based on Melamine-Gallic Acid complex” is a well organized and presented work on the examination of a new melamine complex with the potential to be used as a flame retardant.The main topic that needs to be addressed in my opinion is what was reason for the choice of gallic acid, and whether or not the complexation reaction with melamine provides with enough advantages to be preferred as opposed to a physical mixture of  the two. Melamine is usually combined with phosphates or cyanurates because of their content in phosphorus and nitrogen and subsequent increase of their flame retardation efficiency. While it is known gallic acid can form complexes with melamine, there is no mention of why this was hypothesized to help with its flame retardancy. Additionally, English needs to be edited extensively as the discussion while valid and supported by the results, it gets confusing in many parts of the manuscript. I believe this paper is appropriate for publication after minor revisions that will highlight the significance of the work that was carried out and will increase its scientific soundness.

Specific Comments

  1. Title: Its more common to use the expression fully biobased and not full bio-based. However, its up to the authors if they want to change this.
  2. Abstract: (FTIR, TGA, DSC, DMA, XRD) The Journal’s guidelines to authors state: “Abbreviations should be defined in parentheses the first time they appear in the abstract, main text, and in figure or table captions and used consistently thereafter.”
  3. Introduction page 2 first paragraph: I think it is worth mentioning that biobased PA610, PA1010, PA1012 are available in the market by Evonik, as it proves that the work performed is potentially imminently applicable.
  4. Introduction page 2 first paragraph: What are the applications of such biobased polyamides?
  5. Introduction page 2 second paragraph: The end of the first sentence needs references of the studies about the enhancement of the flame retardancy of PA1010 (if they exist) but also of other biobased PAs.
  6. Introduction page 2 third paragraph: Melamine by itself is available and used as a flame retardant (e.g. AFLAMMIT® PMN 500). What are the advantages of melamine derivatives such as phosphates, cyanurates, oxalate etc.? That might also explain the reason for combining it with GA.
  7. Introduction page 2 fourth paragraph: “The major drawback of Me can be considered its negative impact on the environment, despite its halogen free nature. Forming stable complexes with other molecules can be a route to limit or even restrict this behaviour”. What is the negative impact that melamine has on the environment and what behavior are the authors referring to? How does the formation of complexes help?
  8. Introduction page 2 fourth paragraph: What is GA expected to offer to melamine in comparison with the usual melamine salt components?
  9. Introduction page 3 first paragraph: More flame retardants derived from gallic acid have been reported in the literature (e.g. 10.1007/s11998-019-00273-8, 10.3390/molecules24234305). Their findings could possibly justify the selection of this compounds for this work too.
  10. Materials: Is there a way to produce melamine from renewable sources? If not maybe its more appropriate to remove the phrase “fully biobased” from the title and replace it with partially biobased since the final composite isn’t fully biobased, assuming the Melamine of Sigma Aldrich is produced from a non-sustainable route.
  11. Section 3.1.2: If already obtained, TGA in air would be a valuable addition to the paper as the focus lies on flame retardancy.
  12. Section 3.2.1, first paragraph: In general, this paragraph is not easy to understand and the English must be edited. Also, in the sentence “This effect is expressed by the appearance of bands asymmetrically broadened on the low wavenumber side due to a tendency of antiparallel alignment, observed in the analysed FTIR spectra.”, which bands are the authors referring to in the actual spectra of Figure 4?
  13. Section 3.2.1, page 7, third paragraph: its easier to understand the phrase “overlapping bands” rather than “superimposed absorption bands”.
  14. Section 3.2.2.: same comment with section 3.1.2. Also, can the increase in the residual weights be an indication of improved flame retardancy when the TGA is performed under nitrogen?
  15. Section 3.2.3. Figure 6: Please indicate the direction of endotherms and exotherms in the thermograms.
  16. Section 3.2.3. page 10 last paragraph: the term intercalation is common for clay fillers and its not clear what the authors are trying to explain. Maybe incorporation or dispersion? Is there a plasticization effect that reduces the Tg?
  17. Section 3.2.4.: There is a big difference between the values of the peak of tan delta which is commonly used to calculate the Tg (> 50 °C) with the values calculated by DSC. How can this be explained? The drop in the E’ described as “transition of the material from the glassy to rubbery state” is basically glass transition what occurs in the range 45-70 °C, that is again a lot higher than the values calculated from DSC. It would be helpful if the tan delta peak temperature values were also reported.
  18. Section 3.2.4: The last paragraph is a bit confusing and the use of English language could be improved to make it easily readable. Calculating the area of the tan delta curve could support the conclusions drawn from this section, since increase of this area indicates a great degree of molecular mobility.
  19. Section 3.2.5 second paragraph: correct “tinny”
  20. Section 3.2.5 second paragraph: The figure where the corresponding graphs are presented must be mentioned here.
  21. Section 3.2.5 second paragraph: Are the authors confident that a change in the peak at from 23.7 deg up to 24.12 is considered a “high position difference”? Could it be in the limits of experimental error?
  22. Section 3.2.5 third paragraph: “MA12 provided the most pronounced ability for intercalation in the PA1010 matrix”. Which results indicated  this?
  23. Section 3.2.6: Polymers with LOI > 21 are usually defined as self-extinguishing. According to Figure 10, no sample could be characterized as flammable and the composites could most likely self-extinguish. In my opinion, this is valuable information that should be included in the discussion.
  24. Section 3.2.6: As the difference between the LOI values of PA with MeGA mixture and PA with MeGA complex are very similar (33 and 33.6 respectively), do the authors propose that the complex has significantly better properties as a filler compared to the mixture, or could the complexation procedure be skipped to save time and energy?
  25. Section 3.2.6: Did the authors classify the flammability of the materials with the UL 94 test? It would be interesting and very relevant to this work.
  26. Section 3.2.6: Is this increase in flame retardancy bigger or smaller than what a typical, commercial melamine-based flame retardant usually achieves in polyamides?
  27. Section 3.3: In section 3.2.6 its stated that “no increase in char residue was observed” while here it is stated that “The increase in the TGA residue could suggest the presence of a crosslinking phase”.
  28. Conclusions: This section is way too long. The conclusions in any scientific paper should be a brief, concise and to-the-point section that presents the scope and the originality of the study, briefly present the main and most important findings and limitations, and finally outline possible fruitful lines for further research. In my opinion, it should be maximum half in size.

Author Response

Comments and Suggestions for Authors

The paper “Full bio-based Polyamide 1010 with a halogen-free flame retardant based on Melamine-Gallic Acid complex” is a well organized and presented work on the examination of a new melamine complex with the potential to be used as a flame retardant.The main topic that needs to be addressed in my opinion is what was reason for the choice of gallic acid, and whether or not the complexation reaction with melamine provides with enough advantages to be preferred as opposed to a physical mixture of  the two. Melamine is usually combined with phosphates or cyanurates because of their content in phosphorus and nitrogen and subsequent increase of their flame retardation efficiency. While it is known gallic acid can form complexes with melamine, there is no mention of why this was hypothesized to help with its flame retardancy. Additionally, English needs to be edited extensively as the discussion while valid and supported by the results, it gets confusing in many parts of the manuscript. I believe this paper is appropriate for publication after minor revisions that will highlight the significance of the work that was carried out and will increase its scientific soundness.

 Thank you for the comments and the time spent for the review of our manuscript. We modified accordingly. Yes, indeed, your observation involving the motivation is important to be highlighted. We tried to underline in many places of the manuscript the involvement of the H-bods which promotes the faster thermal decomposition of both Me to ammonia and GA to carbon dioxide and carbon monoxide. By this the action in gaseous form was improved in comparison with neat Me or Me and GA physical mixture. We withdraw the cross-linking mechanism since it was difficult to defend (in agreement with another reviewer).   

Specific Comments

  1. Title: Its more common to use the expression fully biobased and not full bio-based. However, its up to the authors if they want to change this.

We modified accordingly. The term full or fully bio-based was referring to polyamide. We also considered your observation 10 and we changed in just Biobased PA1010.

  1. Abstract: (FTIR, TGA, DSC, DMA, XRD) The Journal’s guidelines to authors state: “Abbreviations should be defined in parentheses the first time they appear in the abstract, main text, and in figure or table captions and used consistently thereafter.”

We modified accordingly.

  1. Introduction page 2 first paragraph: I think it is worth mentioning that biobased PA610, PA1010, PA1012 are available in the market by Evonik, as it proves that the work performed is potentially imminently applicable.

We modified accordingly.

  1. Introduction page 2 first paragraph: What are the applications of such biobased polyamides?

We modified accordingly.

  1. Introduction page 2 second paragraph: The end of the first sentence needs references of the studies about the enhancement of the flame retardancy of PA1010 (if they exist) but also of other biobased PAs.

We modified accordingly.

  1. Introduction page 2 third paragraph: Melamine by itself is available and used as a flame retardant (e.g. AFLAMMIT® PMN 500). What are the advantages of melamine derivatives such as phosphates, cyanurates, oxalate etc.? That might also explain the reason for combining it with GA.

We modified accordingly.

  1. Introduction page 2 fourth paragraph: “The major drawback of Me can be considered its negative impact on the environment, despite its halogen free nature. Forming stable complexes with other molecules can be a route to limit or even restrict this behaviour”. What is the negative impact that melamine has on the environment and what behavior are the authors referring to? How does the formation of complexes help?

We modified accordingly.

  1. Introduction page 2 fourth paragraph: What is GA expected to offer to melamine in comparison with the usual melamine salt components?

We modified accordingly.

  1. Introduction page 3 first paragraph: More flame retardants derived from gallic acid have been reported in the literature (e.g. 10.1007/s11998-019-00273-8, 10.3390/molecules24234305). Their findings could possibly justify the selection of this compounds for this work too.

We modified accordingly.

  1. Materials: Is there a way to produce melamine from renewable sources? If not maybe its more appropriate to remove the phrase “fully biobased” from the title and replace it with partially biobased since the final composite isn’t fully biobased, assuming the Melamine of Sigma Aldrich is produced from a non-sustainable route.

We modified accordingly.

  1. Section 3.1.2: If already obtained, TGA in air would be a valuable addition to the paper as the focus lies on flame retardancy.

We performed the experiments in inert gas (N2) as we found the recommendation in literature. But based on your observation we performed the experiments in air. It was helpful to highlight the flame retardancy.

  1. Section 3.2.1, first paragraph: In general, this paragraph is not easy to understand and the English must be edited. Also, in the sentence “This effect is expressed by the appearance of bands asymmetrically broadened on the low wavenumber side due to a tendency of antiparallel alignment, observed in the analysed FTIR spectra.”, which bands are the authors referring to in the actual spectra of Figure 4?

We modified accordingly. We shortened this section according to another reviewer suggestion.

  1. Section 3.2.1, page 7, third paragraph: its easier to understand the phrase “overlapping bands” rather than “superimposed absorption bands”.

We modified accordingly.

  1. Section 3.2.2.: same comment with section 3.1.2. Also, can the increase in the residual weights be an indication of improved flame retardancy when the TGA is performed under nitrogen?

We modified accordingly, same as for the observation 11.

  1. Section 3.2.3. Figure 6: Please indicate the direction of endotherms and exotherms in the thermograms.

We modified accordingly.

  1. Section 3.2.3. page 10 last paragraph: the term intercalation is common for clay fillers and its not clear what the authors are trying to explain. Maybe incorporation or dispersion? Is there a plasticization effect that reduces the Tg?

We modified accordingly.

  1. Section 3.2.4.: There is a big difference between the values of the peak of tan delta which is commonly used to calculate the Tg (> 50 °C) with the values calculated by DSC. How can this be explained? The drop in the E’ described as “transition of the material from the glassy to rubbery state” is basically glass transition what occurs in the range 45-70 °C, that is again a lot higher than the values calculated from DSC. It would be helpful if the tan delta peak temperature values were also reported.

The difference between the Tg values resulted from DMA (the peak of tan delta) and those obtained from Dsc measurements can be explained by the different heating rates, i.e. 10 °C/min for DSC and 3 °C/min for DMA, which were needed for the experiments. The different heating rate results from the size of the samples. The weight of the DSC sampe was low (around 15mg) with a high surface for the heat transfer, while,by contrast the DMA sample had a much larger weight (around 2.5g) and a small surface area related to sample volume. The data was not obtained for Tg comparison but for highlighting different behaviours between the same set of samples (accordingly with DSC and DMA techniques).

  1. Section 3.2.4: The last paragraph is a bit confusing and the use of English language could be improved to make it easily readable. Calculating the area of the tan delta curve could support the conclusions drawn from this section, since increase of this area indicates a great degree of molecular mobility.

We modified accordingly.

  1. Section 3.2.5 second paragraph: correct “tinny”

We modified accordingly.

  1. Section 3.2.5 second paragraph: The figure where the corresponding graphs are presented must be mentioned here.

We modified accordingly.

  1. Section 3.2.5 second paragraph: Are the authors confident that a change in the peak at from 23.7 deg up to 24.12 is considered a “high position difference”? Could it be in the limits of experimental error?

You are right to have doubts, because is quite specific. We are very confident. The difference is considered a high position difference since in all our polyamide samples with complexes we have a shift (on XRD patterns) (therefore is less probable to be within the experimental errors).  

  1. Section 3.2.5 third paragraph: “MA12 provided the most pronounced ability for intercalation in the PA1010 matrix”. Which results indicated this?

We modified accordingly.

  1. Section 3.2.6: Polymers with LOI > 21 are usually defined as self-extinguishing. According to Figure 10, no sample could be characterized as flammable and the composites could most likely self-extinguish. In my opinion, this is valuable information that should be included in the discussion.

We modified accordingly.

  1. Section 3.2.6: As the difference between the LOI values of PA with MeGA mixture and PA with MeGA complex are very similar (33 and 33.6 respectively), do the authors propose that the complex has significantly better properties as a filler compared to the mixture, or could the complexation procedure be skipped to save time and energy?

We modified accordingly.

MA complexes provide better fire retardant effect, compared with Me and MeGA physical mixture, proven in several aspects of the study (structure, thermo-oxidative degradation or LOI). However the physical mixture MeGA provides close performance to MA complexes, therefore in industrial applications or in the context of balancing with other properties and application, this choice should be considered since it presents evident technological advantages. 

  1. Section 3.2.6: Did the authors classify the flammability of the materials with the UL 94 test? It would be interesting and very relevant to this work.

After your comments we adapted an experimental set-up according to ASTM D3801 and classified the flammability of the materials with the UL-94 test (section 3.2.6).

  1. Section 3.2.6: Is this increase in flame retardancy bigger or smaller than what a typical, commercial melamine-based flame retardant usually achieves in polyamides?

This increase in LOI is a little bigger compared with other melamine-based flame retardant polyamide composites, some examples are presented in the introduction part.

  1. Section 3.3: In section 3.2.6 its stated that “no increase in char residue was observed” while here it is stated that “The increase in the TGA residue could suggest the presence of a crosslinking phase”.

We modified accordingly.

  1. Conclusions: This section is way too long. The conclusions in any scientific paper should be a brief, concise and to-the-point section that presents the scope and the originality of the study, briefly present the main and most important findings and limitations, and finally outline possible fruitful lines for further research. In my opinion, it should be maximum half in size.

We modified accordingly.

Thank you for all the valuable comments and we hope you will find our modifications satisfactory.

Best Regards

Mihai Corobea

Round 2

Reviewer 1 Report

This manuscript can be accepted now.

Author Response

Dear Reviewer 1

In the name of the authors, I thank you for the valuable time spent for improving our manuscript. Thank you for helping us to share our research work.

Best Regards

Mihai Corobea

Reviewer 2 Report

As a reviewer, I suggest that the paper entitled “Bio-based Polyamide 1010 with a halogen-free flame retardant based on Melamine-Gallic Acid complex” should be reconsidered for publication in the Polymers after changes made according to the minor revision.

I propose the following suggestions to the authors, which I believe will improve the - interpretation of the scientific results provided within this manuscript.

  • The abstract should be corrected to be more informative. I suggest to authors to include the explanations for the results (the main conclusions) - In which was MA complexes the PA1010 chain packaging? - In which way decrease of storage modulus influence the composite mechanical properties? - What is the connection between thermal stability thermo-oxidative stability and LOI results and vertical burning behaviour results?
  • Results and Discussion (FTIR – Me, GA and MA complexes): In my previous review report I suggested that the wavelengths, which are commented in the discussion should be marked in the spectra in Figure 2. Please, correct this. (e.g. the wavelengths 3415 cm-1, and 1539 and 808 cm-1 for Me are missing in Figure 4.
  • Results and Discussion (Thermogravimetry – Me, GA and MA complexes): In my previous review report I suggested including results for the Tonset and Tmax temperatures in Table 2. Please, provide and comment the results for the Tonset temperatures (Me, GA and MA complexes), as these temperatures are of high importance for investigating if the starting decomposition temperature of flame retardant matches the starting decomposition temperature of the polymer.
  • Results and Discussion (FTIR – PA1010 and composites): In my previous review report I suggested supporting of the provided discussion by including the results (in this case FTIR peaks). However, there are still some parts in this discussion, which are not supported by the results. Since that the text lines are not numbered, I will copy the text, which should be corrected:
  • Regarding the: “Simultaneously, due to the overlapping bands in a small region, beside the intensity increase, the peaks became sharper” - Please, indicate which bands are overlapping, because otherwise this sentence repeats the information from the previous one.
  • Regarding the: “It was clearly observed that the strongest effect was attained when Me was the major phase in the complex composition.” – According to the spectra presented in Figure 4, the sharpest peak at 3300 cm-1 belongs to the PA10101/MA11 and not to the PA10101/MA21. Please, correct the discussion in accordance to the presented results.
  • Please, correct me if I am wrong, but besides the peaks somewhere at around 1200 and 1100 cm-1, there are not visible peaks at 1166 and 1122 cm-1, which are commented in the discussion.
  • Regarding the: “It was observed that in the lamellar structure the NH-side methylene chains remain in the disordered state a long time after the CO-side methylene chains are parallel arrayed and the intermolecular hydrogen bonds were formed [46]. Our results seem to be in agreement with these findings, as it was already shown in the case of methylene stretching modes” - Please, provide the results, which support this finding.
  • Regarding the: “Therefore, these bands varied in intensity, as it is seen in Figure 4, due to the conformational changes and lateral chain-chain interactions [47], as a result of the perturbations created by the addition of complexes in the amorphous zones, represented by the NH-side methylene segments, as already presented. Especially in the case of “amine-rich” complexes, which are able to interact both with CO and NH groups by hydrogen bonds, the effect of hindering the parallel arrangement of the methylene chains is significantly enhanced, as it will be further confirmed by thermal analysis.” – These comments should also be supported by the results.
  • Results and Discussion (Thermogravimetry – PA1010 and composites): Please, correct the discussion in accordance with the results presented in Figure 5 and Table 4, in which the Tonset temperatures should be included. Namely, according to the results presented in Figs 3 and 5, all the applied FRs (Me, GA, MA11, MA12, MA21) practically completely decompose before PA1010 even start with the decomposition, indicating that the applied flame retardants do not match the pyrolysis specifics of the PA1010. Therefore, even the Me, which has the highest Tonset, among the applied FRs, decomposes completely up to 350 °C, whilst the PA1010 start to decompose at 451 °C. This temperature difference of about 100 °C should not be considered as that Me, GA, and MA complexes decompose just before the start of PA1010 decomposition. The results should be interpreted objectively.
  • Results and Discussion (Thermogravimetry – PA1010 and composites): The thermogravimetry results obtained in air atmosphere should be presented in the same manner as that obtained in nitrogen atmosphere, because the results are equally important as that in nitrogen atmosphere. This refers to graph splitting into two as in the case of nitrogen atmosphere and to including the table with the corresponding Tonset, Tmax temperatures. This also refers to correcting of the currently provided discussion, as the later should include comments regarding the influence of the oxygen from the air environment on the decomposition behaviour of the analysed samples.
  • Results and Discussion (UL94 or ASTM D3801):Please, specify which standard was used for the vertical burning test and include the Table showing the results at least for Tafterflame1, Tfterflame2, Cotton indicator burning (Yes, No), if possible, photographs of the samples after testing.

Author Response

Dear Reviewer 2

Thank you for your time spent and effort in analysing our manuscript. Thank you for contributing to our research visibility and improvement. We made all the modification based on your observation in a purple colour. Below we highlighted also the modifications by each point.

Best Regards

Mihai Corobea  

I propose the following suggestions to the authors, which I believe will improve the - interpretation of the scientific results provided within this manuscript.

  • The abstract should be corrected to be more informative. I suggest to authors to include the explanations for the results (the main conclusions) - In which was MA complexes the PA1010 chain packaging? - In which way decrease of storage modulus influence the composite mechanical properties? - What is the connection between thermal stability thermo-oxidative stability and LOI results and vertical burning behaviour results?

-Indeed and we added on the first review “The MA complexes were formed by hydrogen bonding, starting from 1:2, 1:1, 2:1 Me:GA molar ratios. Since polyamide materials replace in some applications metal materials, improving the flame retardancy could expand the applications of bio-based polyamides. PA1010 composites were obtained by simple methods with industrial relevance (melt mixing, followed by compression moulding).” We added also “The PA1010 MA composites showed a decrease of storage modulus in the dynamic mechanical analysis (DMA).” And “Vertical burning tests (UL-94) showed that all the composites can achieve the UL-94 V-0 rating in contrast with neat PA1010 that has V-2 classification.”

In this second revision based on your observations we added more and we cut out few text lines to fit in the 200 words limit: “This work aims at developing polyamide 1010 (PA1010) composites with improved fire behaviour using a halogen-free flame retardant system based on melamine (Me) and gallic acid (GA) complexes (MA). The MA complexes were formed by hydrogen bonding, starting from 1:2, 1:1, 2:1 Me:GA molar ratios. PA1010 composites were obtained by melt mixing, followed by compression moulding. MA provided a plasticizing effect on the PA1010 matrix, decreasing the glass transition temperature. The influence of MA on PA1010 chain packaging was highlighted in the X-ray diffraction patterns, mainly in the amorphous phase, but affected also the α and γ planes. This was reflected in the dynamic mechanical properties by the reduction of the storage modulus. H-bonds occurrence in MA complexes, improved the efficiency in the gaseous form during fire exposure, facilitating the gas formation and finally reflected in thermal stability, thermo-oxidative stability, LOI results and vertical burning behaviour results. PA1010 containing a higher amount of GA in the complex (MA12) displayed a limiting oxygen index(LOI) value of 33.6 %, much higher when compared to neat PA1010 (25.8 %). Vertical burning tests showed that all the composites can achieve the V-0 rating in contrast with neat PA1010 that has V-2 classification.”

Results and Discussion (FTIR – Me, GA and MA complexes): In my previous review report I suggested that the wavelengths, which are commented in the discussion should be marked in the spectra in Figure 2. Please, correct this. (e.g. the wavelengths 3415 cm-1, and 1539 and 808 cm-1 for Me are missing in Figure 4.

  • We modified accordingly.
  •  
  • Results and Discussion (Thermogravimetry – Me, GA and MA complexes): In my previous review report I suggested including results for the Tonset and Tmax temperatures in Table 2. Please, provide and comment the results for the Tonset temperatures (Me, GA and MA complexes), as these temperatures are of high importance for investigating if the starting decomposition temperature of flame retardant matches the starting decomposition temperature of the polymer.

We modified Figure 2 and added: “Tonset and Tmax temperatures show the starting decomposition temperatures of the FR, earlier than the starting decomposition temperature of the polymer. This behaviour was similar to other polyamide FR’s like Me, but restricts the thermal domain of the potential applications.”

  • Results and Discussion (FTIR – PA1010 and composites): In my previous review report I suggested supporting of the provided discussion by including the results (in this case FTIR peaks). However, there are still some parts in this discussion, which are not supported by the results. Since that the text lines are not numbered, I will copy the text, which should be corrected:
  • Regarding the: “Simultaneously, due to the overlapping bands in a small region, beside the intensity increase, the peaks became sharper” - Please, indicate which bands are overlapping, because otherwise this sentence repeats the information from the previous one.

We modified accordingly and we deleted it.

  • Regarding the: “It was clearly observed that the strongest effect was attained when Me was the major phase in the complex composition.” – According to the spectra presented in Figure 4, the sharpest peak at 3300 cm-1 belongs to the PA10101/MA11 and not to the PA10101/MA21. Please, correct the discussion in accordance to the presented results.

Indeed a big flaw, thank you. We deleted the wrong comment.

  • Please, correct me if I am wrong, but besides the peaks somewhere at around 1200 and 1100 cm-1, there are not visible peaks at 1166 and 1122 cm-1, which are commented in the discussion.

We modified with: In our study, this IR band was situated at 1166 cm-1 and the band specific to the amorphous phase was found at 1122 cm-1. These bands were rather weak therefore an in-depth view on crystallinity should be seen in the XRD section.  

  • Regarding the: “It was observed that in the lamellar structure the NH-side methylene chains remain in the disordered state a long time after the CO-side methylene chains are parallel arrayed and the intermolecular hydrogen bonds were formed [46]. Our results seem to be in agreement with these findings, as it was already shown in the case of methylene stretching modes” - Please, provide the results, which support this finding.

We modified accordingly. We added “...shown in the case of methylene stretching modes in the 2920-2850 cm-1 region.”

  • Regarding the: “Therefore, these bands varied in intensity, as it is seen in Figure 4, due to the conformational changes and lateral chain-chain interactions [47], as a result of the perturbations created by the addition of complexes in the amorphous zones, represented by the NH-side methylene segments, as already presented. Especially in the case of “amine-rich” complexes, which are able to interact both with CO and NH groups by hydrogen bonds, the effect of hindering the parallel arrangement of the methylene chains is significantly enhanced, as it will be further confirmed by thermal analysis.” – These comments should also be supported by the results.

We modified accordingly with “Therefore, these bands varied in intensity, as it is seen in Figure 4 and Table 3, due to the conformational changes and lateral chain-chain interactions [47]. As a result of the perturbations created by the addition of complexes in the amorphous zones (1122cm-1), several small modifications were found for amide I and II region, next to CH2 mode in the 1418-1420 cm-1 region. The effect of hindering the parallel arrangement of the methylene chains from the PA1010 by the MA complexes (summarized in Table 3), was later confirmed in the structure evidenced in the XRD section, in the DMA section by the decrease of the PA1010, modulus and the decrease in Tg seen in DSC section.”

  • Results and Discussion (Thermogravimetry – PA1010 and composites): Please, correct the discussion in accordance with the results presented in Figure 5 and Table 4, in which the Tonset temperatures should be included. Namely, according to the results presented in Figs 3 and 5, all the applied FRs (Me, GA, MA11, MA12, MA21) practically completely decompose before PA1010 even start with the decomposition, indicating that the applied flame retardants do not match the pyrolysis specifics of the PA1010. Therefore, even the Me, which has the highest Tonset, among the applied FRs, decomposes completely up to 350 °C, whilst the PA1010 start to decompose at 451 °C. This temperature difference of about 100 °C should not be considered as that Me, GA, and MA complexes decompose just before the start of PA1010 decomposition. The results should be interpreted objectively.

Indeed another fault, we corrected with: “The results for the thermal decomposition of Me, GA and MA complexes indicate that these compounds decompose much before PA1010 decomposition.

and with: ” All the applied FRs (Me, GA, MA11, MA12, MA21) practically completely decompose before PA1010 even start with the decomposition, indicating that the applied flame retardants do not match the pyrolysis specifics of the PA1010. Therefore, even the Me, which has the highest Tonset, among the applied FRs, decomposes completely up to 350 °C, whilst the PA1010 start to decompose at 451 °C.

and we modified the Figure 5 and Table 4  

  • Results and Discussion (Thermogravimetry – PA1010 and composites): The thermogravimetry results obtained in air atmosphere should be presented in the same manner as that obtained in nitrogen atmosphere, because the results are equally important as that in nitrogen atmosphere. This refers to graph splitting into two as in the case of nitrogen atmosphere and to including the table with the corresponding Tonset, Tmax temperatures. This also refers to correcting of the currently provided discussion, as the later should include comments regarding the influence of the oxygen from the air environment on the decomposition behaviour of the analysed samples.

We modified the Figure 5 and Table 4  and added: The TGA in air underline the involvement of the FR degradation products in the thermo-oxidative PA1010 degradation. These products act in the PA1010 main the thermo-oxidative event (occurring in the 400-480 °C). PA1010 starts the main event in air at 420 °C (Figure 5d Deriv.Weight), were like any aliphatic polyamide, the abstraction of the hydrogen atom from N-vicinal methylene group occurs. The second maxima of PA1010 involved the propagation involved by the oxidation of the formed macroradical. The FR degradation products suppress the first event from 420 °C by enriching the PA1010 decomposition loci with gaseous species. The second event of PA1010 thermo-oxidative degradation (oxidation of the macroradical) was influenced also especially by the MA type of FR (Figure 5d Deriv.Weight), creating a different mechanism with third phase more evident near 500°C . The FR suppression of the oxidative events was in good agreement with LOI measurements.

  • Results and Discussion (UL94 or ASTM D3801):Please, specify which standard was used for the vertical burning test and include the Table showing the results at least for Tafterflame1, Tfterflame2, Cotton indicator burning (Yes, No), if possible, photographs of the samples after testing.

We modified accordingly (ASTM D3801). Unfortunately, the photos were not available.